

# Tropospheric Ozone Assessment Report (TOAR): 16-year ozone trends from the IASI Climate Data Record

Anne Boynard[1,2], Catherine Wespes[3], Juliette Hadji-Lazaro[1], Selviga Sinnathamby[1], Daniel Hurtmans[3], Pierre-François Coheur[3], Marie Doutriaux-Boucher[4], Jacobus Onderwaater[4], Wolfgang Steinbrecht[5] Elyse A. Pennington[6], Kevin Bowman[6], and Cathy Clerbaux[1,3]

[1]LATMOS/IPSL, Sorbonne Université, UVSQ, CNRS, Paris, France
[2]SPASCIA, Ramonville-Saint-Agne, France
[3]Spectroscopy, Quantum Chemistry and Atmospheric Remote Sensing, Université Libre de Bruxelles (ULB), Brussels, Belgium
[4]EUMETSAT, Darmstadt, Germany
[5]Deutscher Wetterdienst, Hohenpeissenberg, Germany
[6]Jet Propulsion Laboratory, Pasadena, California, United States of America

*Correspondence to*: Anne Boynard (anne.boynard@latmos.ipsl.fr)

**Abstract.** Assessing tropospheric ozone ($O_3$) variability is essential for understanding its impact on air quality, health, and climate change. The Infrared (IR) Atmospheric Sounding Interferometer (IASI) mission onboard the Metop platforms, has been providing global measurements of $O_3$ concentrations since 2007. This study presents the first comprehensive analysis of the 16-year $O_3$ Climate Data Record (CDR) from IASI/Metop (2008–2023), a homogeneous dataset offering valuable insights into the variability and long-term trends of tropospheric $O_3$. The IASI-CDR ozone product is evaluated against TROPESS (TRopospheric Ozone and its Precursors from Earth System Sounding) $O_3$ retrievals from the Cross-track Infrared Sounder (CrIS). The comparison shows excellent agreement for total ozone (biases < 1.2%, correlations > 0.97) and good agreement for tropospheric ozone (biases 10–12%, correlations 0.77–0.91). Comparisons with ozonesonde data indicate that IASI underestimates tropospheric ozone by 2% in the tropics and by up to 10% in mid and high latitudes. Spatiotemporal analysis of IASI data from 2008 to 2023 reveals a global negative trend in tropospheric $O_3$ (-0.40 ± 0.10% year$^{-1}$), with the most pronounced decreases observed in the tropics and in Europe. Despite differing from positive trends in ultraviolet (UV) satellite data, both UV and IR satellite instruments show a significant drop in tropospheric ozone starting in 2020, partly due to pandemic-related emission reductions. This study emphasizes the importance of long-term, consistent datasets for tracking ozone trends and the need for improved data retrieval and integration to address regional and temporal discrepancies.

## 1 Introduction

Tropospheric and stratospheric ozone ($O_3$) are key components of Earth's atmosphere with distinct roles and impacts. Stratospheric ozone plays a protective role by absorbing the majority of the Sun's harmful ultraviolet (UV) radiation, thus shielding life on Earth. The depletion of stratospheric ozone due to chlorofluorocarbons (CFCs) has led to global efforts like



the Montreal Protocol, which has been essential in mitigating further losses (World Meteorological Organization, 2018). The interactions between the two layers are complex, as stratosphere-troposphere exchange processes influence tropospheric

ozone levels and, subsequently, surface air quality (Neu et al., 2014; Ziemke et al., 2019; Thompson et al., 2021).

In contrast, tropospheric ozone is a secondary pollutant formed through photochemical reactions involving precursors such as nitrogen oxides ($NO_x$), volatile organic compounds (VOCs) and carbon monoxide (CO) in the presence of sunlight (Seinfeld and Pandis, 1998). It is a short-lived climate forcer and a major contributor to air pollution, affecting human health by aggravating respiratory and cardiovascular conditions and reducing crop yields and ecosystem productivity (Fleming et

al., 2018; Mills et al., 2018; Tai et al, 2014; Murray et al, 2024). Furthermore, both tropospheric and stratospheric ozone play a significant role in radiative forcing, with tropospheric ozone having particularly pronounced effects in tropical and subtropical regions (Worden et al, 2008; Bowman and Henze, 2012; Gaudel et al., 2018). By absorbing infrared radiation, it also contributes to global warming, acting as a potent greenhouse (IPCC, 2021). Understanding its sources and dynamics is essential for developing effective mitigation strategies to address its environmental and health impacts.

Recent studies have significantly advanced our understanding of tropospheric ozone trends, particularly those derived from satellite data, emphasizing the complexity and variability of ozone concentrations across different regions and time scales. These trends are influenced by both natural and anthropogenic factors (Lelieveld et al., 2008; Wespes et al., 2017; Tarasick et al., 2019; Ziemke et al., 2019). For example, the first phase of the Tropospheric Ozone Assessment Report (TOAR-I) highlighted that free tropospheric ozone has increased since the industrial era and continued to rise in recent decades (Gaudel

et al., 2018; Tarasick et al., 2019; Gulev et al., 2021). Regional trends are more varied: while ozone levels have decreased in summer in North America and Europe, they have increased in Asia (Gaudel et al., 2018, 2020; Wespes et al., 2018). Several studies based on in situ measurements have observed similar patterns. Stauffer et al. (2024) analyzed 25 years of ozone profiles over equatorial Southeast Asia, observing increases in free tropospheric ozone (5–15% per decade), which they attributed to reduced convection between February and April, suppressing ozone dilution and promoting the accumulation of

biomass burning emissions. Likewise, Wang et al. (2022) reported rising global ozone levels, especially in the free troposphere, from 1980 to 2017 using IAGOS data. In both cases, limitations in spatial sampling have to be considered (Miyazaki and Bowman, 2017).

The COVID-19 pandemic in 2020 marked a significant turning point in tropospheric ozone trends. Studies indicate that reduced emissions of ozone precursors during the economic slowdown of 2020 led to declines in ozone levels (Miyazaki et

al., 2020; Ziemke et al., 2021). Satellite-based observations suggest that the increase of the tropospheric ozone burden ended in 2020, with no increase observed between 2021 and 2023 (Dunn et al., 2024). This stagnation coincided with negative ozone anomalies in the free troposphere in the Northern Hemisphere (NH) in 2020, documented by ozonesonde, lidar and commercial aircraft data (e.g. Steinbrecht et al., 2021; Chang et al., 2022, 2023a). Model simulations attribute these decreases to reduced emissions of ozone precursors, with levels comparable to those of the mid-1990s (Miyazaki et al.,

2020; Steinbrecht et al., 2021).



Despite these advancements, reconciling trends derived from satellite data remains challenging due to discrepancies between UV and infrared (IR) satellite sounders (Gaudel et al., 2018). UV sounders generally report positive trends (e.g., Cooper et al., 2014; Ziemke et al., 2019; Liu et al., 2022; Fadnavis et al., 2024; Gaudel et al., 2024), while IR sounders often indicate negative trends (Wespes et al., 2018; Dufour et al., 2018, 2021, 2024). However, more recent studies have begun to bridge these differences. For instance, Pope et al. (2024) identified weak negative trends with significant uncertainties in lower tropospheric ozone over North America, Europe, and East Asia from 2008 to 2017 using both UV and IR data. Similarly, Pimlott et al. (2024) reported small negative trends in European lower tropospheric ozone during the same period based on UV satellite observations. These findings underscore the variability of trends depending on the temporal period and highlight the importance of harmonizing and validating satellite observations to improve the reliability of ozone trend analyses.

Further investigation is needed to enhance our understanding of tropospheric ozone variability, improve knowledge of ozone trends, and ensure consistent long-term monitoring. The Infrared Atmospheric Sounding Interferometer (IASI) instruments onboard the Metop satellites provide a consistent dataset since 2007, enabling long-term ozone monitoring. This study, conducted within the framework of the TOAR-II project, analyzes the IASI $O_3$ Climate Data Record (IASI-CDR), recently processed by EUMETSAT under the auspices of the Atmospheric Composition Monitoring Satellite Application Facility (AC SAF) project, offering a homogeneous 16-year dataset (2008-2023) for the first time. The primary objectives are (1) to assess the quality and consistency of the IASI-CDR $O_3$ product, and (2) to investigate the spatiotemporal variability and long-term ozone trends. This work aims to deepen our understanding of ozone dynamics and its evolution over time. Section 2 describes the datasets and the methods used in this study, Section 3 presents the validation results of the IASI-CDR $O_3$ product using independent measurements, Section 4 gives a comprehensive analysis of tropospheric ozone trend estimates, and Section 5 summarizes the main findings of this study.

## 2. Data and methods

### 2.1 Data

#### 2.1.1 The IASI-CDR $O_3$ product

The IASI instruments are carried by the Metop satellites, launched in 2006 (Metop-A), 2012 (Metop-B), and 2018 (Metop-C) (Clerbaux et al., 2009). Metop-A entered orbital drift in 2017 and stopped providing IASI data in October 2021. The IASI instruments provide data twice daily with overpass times at approximately 9:30 AM and 9:30 PM local solar time, and spatial resolution of 12 km diameter at nadir. The instrument has a spectral resolution of 0.5 cm⁻¹ (after apodization) over the 645–2760 cm⁻¹ range and a radiometric resolution of 0.25 K for temperature at 280 K (Hilton et al., 2012), ensuring high precision for atmospheric composition monitoring.



Various research groups have developed $O_3$ retrieval algorithms for IASI using different approaches (e.g., Barret et al., 2020; Dufour et al., 2012; Hurtmans et al., 2012). In particular, ULB and LATMOS developed the Fast Optimal Retrievals on Layers for IASI (FORLI) software (Hurtmans et al., 2012), which uses IASI Level 1C and meteorological Level 2 data to retrieve $O_3$ Level 2 products. The IASI-FORLI $O_3$ product has been widely used in numerous studies such as urban pollution (Ancellet et al., 2024), ozone hole monitoring (Scannell et al., 2012; Gazeaux et al., 2013; Safieddine et al., 2020),

interannual variability (Safieddine et al., 2013, 2014, 2016; Wespes et al., 2016, 2017) and long-term trend analyses (Gaudel et al., 2018; Wespes et al., 2018; Pope et al., 2024). It has also been extensively validated (Dufour et al., 2012; Pommier et al., 2012; Boynard et al., 2016, 2018; Keppens et al., 2018). In particular, Boynard et al. (2018) validated the latest version of IASI-FORLI v20151001 $O_3$ product for the period over 2008-2017, reporting a global mean difference of less than 2% in the total ozone column (TOC) compared to independent measurements such as GOME-2, with larger discrepancies observed

at high latitudes. For the tropospheric ozone column, they found a positive bias of 4–5% at high latitudes and a negative bias of 11–19% in midlatitudes and the tropics.

A few years ago, the FORLI v20151001 version was implemented in the EUMETSAT near real time (NRT) processing facility in the framework of the AC SAF project. Since 4 December 2019, the IASI-FORLI $O_3$ product has been operationally distributed to users through the EUMETCast system and made available to users on the AERIS website

(iasi.aeris-data.fr/O3). The IASI-FORLI $O_3$ Level 2 and Level 3 products are included in the European Space Agency Ozone Climate Change Initiative (Ozone_cci, http://www.esa-ozone-cci.org) and the European Centre for Medium Range Weather Forecasts (ECMWF) Copernicus Climate Change (C3S, https://cds.climate.copernicus.eu/) projects, respectively. These initiatives aim to develop comprehensive $O_3$ datasets that are relevant for climate monitoring, specifically as essential climate variables (ECVs). However, the IASI-FORLI dataset is not consistent, as it was processed based on different

versions of the IASI Level 1C (radiances) and Level 2 (temperature, humidity and cloud) Product Processing Facility (PPF) between 2007 and 2019, as summarized in Tables 1 and 2 of Bouillon et al. (2020). Consequently, it should be used with caution for long-term trend studies, as these differences have been shown to introduce a "jump" leading to an "artificial drift" in the tropospheric ozone time series, particularly around the end of 2010 (Boynard et al., 2018).

In this context, EUMETSAT AC SAF has reprocessed the IASI-FORLI v20151001 product using the reprocessed IASI

Level 1C radiances (EUMETSAT, 2018) and Level 2 temperature and humidity products (Doutriaux-Boucher and August, 2020), to ensure consistency and homogeneity throughout the entire IASI period. This process resulted in the creation of a homogeneous IASI $O_3$ product: the IASI-CDR $O_3$ product. To account for potential biases arising from the orbital drift of Metop-A, only IASI-A data up to December 2019 are included in the CDR. From that point onwards, the record seamlessly continues using IASI-B data, as the two instruments are identical (Boynard et al., 2018; Bouillon et al., 2020). Additionally,

Metop-B has been reprocessed for 2019 to provide a common year with IASI-A. Due to the computational demands of the ozone product, IASI-B data for the period overlapping with IASI-A (2013–2018) were not reprocessed. Data from IASI-C, which is the same instrument as IASI-A and IASI-B, are not reprocessed into the CDR but are produced using the same version of EUMETSAT Level 1C and 2 PPF and FORLI algorithm (v20151001) as the CDR, ensuring consistency and



forming the Interim CDR (ICDR) component. Henceforth, we will refer to two IASI products: the inhomogeneous IASI-
FORLI product and the homogeneous IASI-CDR product.

The IASI-CDR $O_3$ dataset analyzed in this study covers the period January 2008 to December 2023 (2008–2019: IASI-A; 2020–2023: IASI-B+C), with only 3.5 out of 10 IASI pixels processed due to computing time constraints. This dataset provides vertical ozone profiles as partial columns distributed across 40 atmospheric layers (from the surface to 40 km altitude), along with a separate column for the upper atmosphere above 40 km. Each retrieval includes quality metrics such

as a priori profiles, total error estimates, averaging kernel (AK) matrices, and quality flags. To ensure data quality, only retrievals associated with a general quality flag (GQF) equal to 1 are used. This GQF includes a cloud flag to filter out cloud-contaminated scenes, allowing retrievals to be performed only on clear or mostly clear scenes with fractional cloud cover below 13%, as identified using cloud information from the EUMETSAT operational processing (August et al., 2012). It is also recommended to exclude data associated with low degrees of freedom for the signal (DOFS < 2), a limitation often

affecting data from polar regions like Antarctica (Boynard et al., 2018). In this study, we analyze both total and tropospheric ozone columns. For the tropospheric column, we use the column integrated from the surface to the thermal tropopause, as defined by the World Meteorological Organization (WMO) (WMO, 1957). The thermal tropopause is calculated from the IASI L2 temperature profiles. As highlighted by Elshorbany et al. (2024), relying on a fixed pressure level for the tropopause may lead to inaccuracies due to temporal variations in the thermal tropopause height, which justifies the use of the thermal

tropopause in this analysis.

Figure 1 shows the vertical sensitivity of IASI in the polar, midlatitude, and tropical regions, separated into day and night conditions. In all regions, the averaging kernel (AK) values do not reach their maximum at the nominal altitudes, indicating that the ozone retrieval at each level is impacted by contributions from other altitudes. The maximum sensitivity is observed around 8 km, regardless of day or night conditions. For the midlatitude and tropical regions, there are no significant

differences between day and night sensitivity. However, at high latitudes, the sensitivity in the lower troposphere is notably weaker at night, as shown by both the averaging kernel curves and the DOFS values (DOFS of ~3 during the day and ~2.5 at night). The tropical regions, benefiting from higher temperatures, show the highest sensitivity, with DOFS values of ~4.

The variability in data sensitivity and quality is reflected in both the error and DOFS analyses, revealing differences across latitude bands and between day and night (see Table 1). In high latitude regions (60°-90°N and 60°-90°S), total column

errors are significantly higher, particularly in the Southern Hemisphere (SH), where daytime errors reach 4.43% compared to 3.64% at night. Similarly, in the NH, daytime errors (~1.37%) are higher than nighttime errors (~0.95%). Tropospheric column errors are even more pronounced in these regions, especially in the SH, where daytime errors peak at ~19.91%. These results highlight reduced data quality and sensitivity in polar regions, particularly at night, due to lower surface temperatures and weaker signals. In the midlatitudes (30°-60°N and 30°-60°S), total column errors are relatively low and

consistent between day and night, ranging from ~0.95% to ~1.24%. In the tropics (0°-30°N and 0°-30°S), total column errors are even lower, ranging from ~0.90% to ~1.00%. Tropospheric column errors in these regions are similarly low, with slightly higher values in the SH midlatitudes, suggesting generally higher data quality. Tropical regions benefit from more favorable



atmospheric conditions and higher temperatures, which contribute to significantly better data quality compared to the polar regions.

The DOFS analysis provides further insights into data sensitivity. In high-latitude regions, the DOFS values for the total column decrease at night (e.g., ~2.85 during the day vs. ~2.54 at night in the NH). Tropospheric DOFS values are particularly low in the SH (~0.32 during the day and ~0.43 at night), reflecting limited sensitivity and reduced information content in polar regions. In the midlatitudes, total column DOFS values are higher but show a slight decrease from day to night (e.g., ~3.29 during the day vs. ~3.15 at night in the NH). Tropospheric DOFS remain around ~1.0, indicating moderate

sensitivity. In the tropics, DOFS values are the highest for both the total and tropospheric columns. Total column DOFS remain consistent between day and night (~3.7), while tropospheric DOFS are around ~1.7, reflecting enhanced sensitivity and information content in these regions.

In summary, the tropics demonstrate the highest data quality and sensitivity, while the polar regions, particularly at night, show reduced sensitivity and increased uncertainty, especially in the tropospheric column. This pattern reflects the influence

of atmospheric conditions and instrument performance across regions and times.

To analyze tropospheric ozone, it is essential to first validate the total ozone column, followed by the tropospheric ozone column. In the following, we describe both the total and tropospheric columns, which will be validated in Section 3.

Figure 2, which illustrates the spatial and seasonal variability of total ozone from the IASI-CDR product, shows a pronounced latitudinal gradient. Ozone concentrations are lower in the tropics, where the ozone layer is thinner, and increase

toward higher latitudes. This pattern can be observed each year, as shown in Fig. 3 illustrating the monthly time series of the IASI $O_3$ total columns between 2008 and 2023. This is primarily driven by two factors: significant ozone production in the tropics due to intense solar UV radiation and the Brewer-Dobson circulation, which transports ozone from the tropics to the poles, leading to an accumulation at middle and high latitudes (Dessler, 2000; Wespes et al., 2018). Seasonal variations also play a role. At high latitudes, ozone levels are lowest in winter due to the polar night and the isolation of ozone-poor air by

the polar vortex, while levels increase in spring and early summer as sunlight returns. However, polar regions face to lower levels in late summer due to photochemical destruction. We clearly see very low levels of total ozone in Antarctic, where photochemical reactions during the austral spring cause the ozone hole (WMO, 2018). Figure 3 also highlights key events, such as the largest ozone hole in 2015 (associated with El Niño), the smallest ozone hole in 2019 (linked to a sudden stratospheric warming; Safieddine et al. (2020)) over Antarctica, and a smaller ozone hole in 2020 over the North Pole. At

midlatitudes, total ozone peaks in spring when ozone is transported from the tropics and decreases in summer due to photochemical loss, with lower levels in fall and winter. In contrast, the tropics have relatively stable ozone levels throughout the year, as sunlight and ozone transport remain relatively stable. Another important insight provided in Fig. 3 is the comparison of the total ozone columns across the three IASI instruments and two IASI products (IASI-CDR and IASI-FORLI). This will be discussed hereafter.

Figure 4 (left panels) illustrates the spatial and seasonal variability of tropospheric ozone from the IASI-CDR product. Tropical regions have lower ozone levels due to humid conditions and active convection, with localized plumes during



biomass burning seasons. The midlatitudes experience higher concentrations in spring and summer, driven by photochemical production from anthropogenic emissions of precursors like $NO_x$ and VOCs, particularly over densely populated areas of China and northern India (Logan, 1985; Fusco and Logan, 2003; Safieddine et al., 2013; Wespes et al., 2016). Long-range transport, such as ozone from Asia to the Pacific, also contributes to variability (Zhang et al., 2008). In the eastern Mediterranean Basin, summer ozone levels are linked to stratosphere-troposphere exchange (Safieddine et al., 2014). Significant fire activity around 20°S–40°S in spring also affects seasonal distributions (Wespes et al., 2017). High latitudes have lower background levels due to minimal emissions and reduced photochemistry, although enhancements occur in spring, influenced by transport and stratosphere-troposphere exchange, with additional summer increases in the Arctic (Stohl et al., 2003). Winter months show reduced photochemical activity across all regions due to lower sunlight. These patterns are observed annually, as shown in Fig. 5, which illustrates the monthly time series of tropospheric ozone from the IASI-CDR product.

As shown in Fig. 4 (middle panels), the standard deviation for tropospheric ozone is larger in polar region, which is due to the lower IASI sensitivity in those regions because of low temperature. The larger standard deviation observed during the June-July-August (JJA) season over Antarctica is partly due to the complexities involved in accurately defining the tropopause altitude in this region. Unlike other areas, Antarctica can exhibit two distinct tropopauses within the temperature profile, making the conventional temperature-based definition of the tropopause unreliable. This phenomenon complicates the identification of a single tropopause, which can lead to discrepancies in understanding the distribution of ozone and the exchange between the troposphere and stratosphere. The use of temperature profiles to estimate tropopause height is thus not suitable for the South Pole, where alternative dynamic or chemical criteria are more appropriate for defining the tropopause structure (Zängl et al. 2001; Xian et al., 2019).

The DOFS associated with the tropospheric ozone column, shown in Fig. 4 (right panels), exhibit a clear meridional gradient, with higher values in the tropics and lower values in the polar regions. This gradient is driven by the temperature-dependent IASI sensitivity, which influences the amount of information that can be retrieved across different latitudes. In the tropics, elevated temperatures enhance IASI sensitivity, resulting in consistently high DOFS values near 2.0 across all seasons, indicating that two independent pieces of information can be reliably retrieved. In contrast, midlatitude regions show more variability in DOFS, typically ranging from 1.0 to 1.5, due to more fluctuating temperatures. Polar regions, where IASI sensitivity is reduced due to colder temperatures, generally exhibit lower DOFS values, often falling below 1.0. The seasonal patterns further emphasize these differences: the tropics maintain consistent DOFS values year-round, while midlatitudes experience greater fluctuations, particularly during transitional seasons like spring and fall.

Figures 3 and 5 compare the monthly total and tropospheric ozone time series, respectively, between the three IASI instruments with both IASI-CDR and IASI-FORLI products. A strong consistency is found among the three IASI instruments, with a global bias of less than 0.05% for the total column and less than 1% for the tropospheric column. This consistency is observed between Metop-B and -C from 2020 to 2023, and between Metop-A and -B for 2019, as these are the only overlapping periods for these instrument pairs in the IASI-CDR product. For the total column (see Fig. 3), there is a





strong agreement between IASI-CDR and IASI-FORLI for Metop-A and -B, which suggests that the total column product is not impacted by the different changes in Eumetsat L2 PPF. However, for the tropospheric column (see Fig. 5), large differences between the two products are observed until end of 2010, corresponding to a change in the Eumetsat Level 2 PPF version. After this period, agreement improves notably across all latitude bands although discrepancies remain visible
especially in the high latitudes and southern midlatitudes. Our analysis of the monthly ozone time series shows that IASI-CDR and IASI-FORLI converge starting from late 2019, as expected, since the same version of Eumetsat L2 PPF was used for both products.

### 2.1.2 The TROPESS-CrIS O$_3$ product

The Cross-track Infrared Sounder (CrIS) is an infrared Fourier transform spectrometer on board the NOAA Suomi- National
Polar-orbiting Partnership (Suomi-NPP) and the Joint Polar Satellite System1 (JPSS-1 or NOAA-20) satellite operating since 2011 and 2017, respectively (https://www.jpss.noaa.gov/mission_and_instruments.html, and https://www.star.nesdis.noaa.gov/jpss/CrIS.php) (Luo et al., 2024). CrIS measures radiances at over 1300 channels with a spectral resolution of 0.625 cm$^{-1}$ in three spectral bands—long-wave infrared (650–1095 cm$^{-1}$), mid-wave infrared (MWIR) (1210–1750 cm$^{-1}$), and short-wave infrared (2155–2550 cm$^{-1}$). It achieves near-global coverage twice daily with overpass
times at approximately 1:30 AM and 1:30 PM local solar time, different from that of IASI/Metop. Each CrIS pixel or field of view (FOV) is circular with a 14 km radius at nadir (Luo et al., 2024; Malina et al., 2024).
We use the TROPESS (TRopospheric Ozone and its Precursors from Earth System Sounding) CrIS ozone reanalysis stream summary product (TROPESS-CrIS, 2023), which contains vertically resolved concentrations of atmospheric O$_3$. Leveraging the MUlti-SpEctra, MUlti-SpEcies, Multi-Sensors (MUSES) algorithm (Fu et al., 2016), derived from the heritage of
Aura/TES, TROPESS provides global datasets for the time period from December 2015 to December 2022. In late March 2019, an anomaly in the MWIR band caused a data gap from April to July 2019, which may affect trend analyses. After the instrument was fully restored, the data from early and late 2019 showed strong consistency (Iturbide-Sanchez et al., 2022). TROPESS uses an optimal estimation retrieval approach (Rodgers, 2000), building on the level-2 processing framework developed for Aura/TES (Tropospheric Emission Spectrometer) by Bowman et al. (2006). The MUSES algorithm retrieves
atmospheric temperature, water vapor, and other trace gases simultaneously with ozone in the presence of clouds. The data are reported at 26 vertical levels from the surface to 0.1 hPa. TROPESS-CrIS O$_3$ products have been used in previous scientific studies; for example, tropospheric ozone changes during the COVID-19 lockdown have been assessed using CrIS data (Miyazaki et al., 2021); ozone retrieval improvement has been assessed by combining measurements from CrIS and from the TROPOspheric Monitoring Instrument (TROPOMI) aboard the Copernicus Sentinel-5 Precursor (S-5P) satellite
(Melina et al., 2024). Pennington et al. (2024) compared the TROPESS-CrIS O3 products with ozonesonde data and found a stable evolution of global tropospheric ozone bias of $0.21 \pm 3.6$ % decade$^{-1}$ for the period 2016–2021.





In this study we analyze the TROPESS-CrIS O₃ product over the period from January 2016 till December 2022. The data are averaged daily on a global grid with a resolution of 1°x1°. Figures 6 and 7 illustrate the spatial and seasonal distribution of CrIS O₃ total and tropospheric column, respectively, over the period 2016-2022. The spatial and season patterns between

IASI and CrIS data are in very good agreement for both columns. A detailed comparison of these datasets is conducted in Section 3.

### 2.1.3 Ozonesonde data

We use ozonesonde data from the TOAR-II project, as part of the Harmonization and Evaluation of Ground-based Instruments for Free-Tropospheric Ozone Measurements (HEGIFTOM) working group (HEGIFTOM, 2025), which have

been corrected for possible biases related to instrumental or processing changes (Van Malderen et al., 2025). The incomplete HEGIFTOM sonde time series are supplemented with data from the World Ozone and Ultraviolet Radiation Data Centre (WOUDC, 2025). Additional ozonesonde data were acquired from NOAA (NOAA, 2025), the Southern Hemisphere Additional Ozonesondes (SHADOZ; 2025). Except Hohenpeissenberg which uses the Brewer-Mast type, all sites use electrochemical concentration cell (ECC) sensors, which offer high accuracy (3%–5%) and provide reliable measurements of

ozone from the surface up to around 35 km altitude, with a vertical resolution of about 100 m (Tarasick et al., 2021). The precision of these measurements varies between 5%–15% in the troposphere and around 5% in the stratosphere (Sterling et al., 2018; Witte et al., 2018; Steinbrecht et al., 2021).

A total of 43 ozonesonde stations across midlatitudes, polar, and tropical regions, covering the period from 2008 to 2023, were considered in this study. HEGIFTOM harmonized data were available for 29 of these sites at the time of the study, and

incomplete time series were filled using WOUDC data. Additionally, data from 14 more ozonesonde sites were supplemented by NOAA/SHADOZ. Figure 8 shows the locations of the ozonesonde stations used in the comparison and Table 2 described the station characteristics.

### 2.2 Comparison methodology

Given the differences in dataset characteristics, this section outlines the comparison methodologies, collocation criteria, and

statistical analysis.

### 2.2.1 Comparison with CrIS data

Comparing IASI and CrIS is challenging due to differences in pixel localization in both time and space. A simple way to compare collocated data is by computing the daily average of IASI and CrIS over a constant $1° \times 1°$ grid cell. The monthly averages are calculated using the daily gridded averages. Both daytime and nighttime data are included in the comparison.

The differences are obtained by: [IASI − CrIS] (in DU) or [100 × (IASI − CrIS) / (0.5 x (IASI + CrIS))] (in percent, %), where the average of CrIS and IASI serves as a combined reference for the comparison.



The comparison between IASI and CrIS is performed for the total and tropospheric columns over the common period 2016-2022.

### 2.2.2 Comparison with sonde data

Prefiltering is applied by excluding soundings that do not reach 80 hPa and those with total ozone correction factors outside the range of [0.7, 2.0]. Sonde data are aggregated into 46 pressure layers, ranging from 900 hPa to 8 hPa, with values representing vertical averages. The averaging is performed using the midpoints between the given pressure levels.

For the comparison of IASI data with sonde measurements, profiles are paired when the sonde is launched within 1° and 6 hours of the IASI measurement. The chosen thresholds ensure sufficient data for statistically meaningful comparisons and

are consistent with methodologies used in previous studies (e.g., Boynard et al., 2016, 2018). These criteria can lead to multiple IASI profiles being matched to a single sonde profile.

For the validation of satellite profile products, the difference in vertical resolution has to be taken into account. In this study, for each IASI/sonde pair, the sonde profiles are interpolated onto the IASI vertical grid and degraded to the IASI vertical resolution by applying the IASI AKs and a priori $O_3$ profile according to Rodgers (2000):

$x_s = x_a + \mathbf{A}(x_{raw} - x_a)$  (Eq. 1)

where $x_s$ is the smoothed ozonesonde profile, $x_{raw}$ is the ozonesonde profile interpolated on the IASI vertical grid, $x_a$ represents the IASI a priori profile and $\mathbf{A}$ refers to the IASI AK matrix. The sonde profiles above the burst altitude are extended using the IASI a priori profile.

For each sonde measurement, we compute the tropospheric column by using all IASI and smoothed sonde profiles that meet

the coincidence criteria. Then, the IASI and smoothed sonde subcolumns are averaged, yielding one IASI-SONDE profile pair corresponding to each sonde measurement. Both IASI $O_3$ vertical profiles and tropospheric columns are compared against ozonesonde data. The differences are determined as: [IASI − SONDE] (in DU) or [100 × (IASI − SONDE) / SONDE] (in %), where SONDE serves as a reference for the comparison. We also assess the four subcolumns defined in Boynard et al. (2018), each containing approximately one piece of information and showing maximum sensitivity around the

middle of each layer. The subcolumns correspond to the following layers: surface–300 hPa, 300–150 hPa, 150–25 hPa and 25–3 hPa. The results are presented in Appendix A.

### 2.3 Trend evaluation

To calculate trends, the monthly ozone column time series are first deseasonalized to eliminate seasonal variations. Trend analysis is then performed using the quantile regression (QR) method, focusing on the median (50th percentile). QR is

preferred over other trend detection methods for its robustness against small sample sizes, resilience to outliers, and ability to accommodate non-normal error distributions and autocorrelation. Trend calculations follow the guidelines set by the TOAR-II statistics focus working group (Chang et al., 2023b). The TOAR-II methodology uses QR at the 50th percentile for trend



estimation, along with a moving block bootstrap approach to assess trend uncertainty and calculate p-values to evaluate the likelihood of a trend. This bootstrapping method calculates the standard error of the trend across multiple subsamples (or

"blocks") of the time series, effectively accounting for autocorrelation and heteroskedasticity. Uncertainties are estimated at the 95% confidence level using block bootstrapping. Global trend calculation from IASI-CDR O$_3$ data is carried out on a 1° × 1° grid, considering only regions with at least 70% of the monthly values available over the 2008–2023 period.

For the local trend calculation at individual sonde station, we use a different approach. The sampling frequency, typically fewer than 2–4 profiles per week, poses challenges for trend estimation, as robust trend analyses require higher temporal

resolution (Chang et al., 2020). To address these limitations, for trend calculation, the following selection criteria adapted from Lu et al. (2019) are applied: (1) at least three observations per month to calculate monthly means and (2) a minimum of 70% of the monthly time series over the 2008–2023 period. Reducing the number of measurements per month (to 2 or 1) and/or the proportion of non-NaN monthly values (e.g., 60%, 50%) alters the observed trends.

The data requirements for this analysis are met by seven ozonesonde stations over 43 for the period from 2008 to 2023 (see

station names in bold in Table 1). Four of these stations are located in Europe (Uccle, Payerne, Hohenpeissenberg and Madrid), one in North America (Boulder), one in the tropics (Hilo), and one in the southern midlatitudes (Lauder). Due to the distribution of the stations, the global trends assessed are mainly representative of Europe. Only a limited number of stations meet the required criteria, primarily due to the fact that not all IASI pixels have been processed (only 3.5 out of 10). This processing gap reduces the overlap between IASI data and sonde measurements. Additionally, the availability of sonde

stations is limited, with most providing fewer than three measurements per month, further restricting the potential for meaningful coincidences.

According to the TOAR-II statistical guidance of Chang et al. (2023b), trends have been classified by significance as follows: p-value ≤ 0.01 (very high certainty), 0.01 < p ≤ 0.05 (high certainty), 0.05 < p ≤ 0.10 (medium certainty), 0.10 < p ≤ 0.33 (low certainty) and p > 0.33 (very low certainty or no evidence).

**3. Validation**

**3.1 Comparison with CrIS data**

Seasonal comparisons between IASI-CDR and CrIS-TROPESS products are conducted to evaluate their performance for two ozone columns: total and tropospheric. The analysis includes scatterplots and spatial distribution maps, highlighting both the agreements and discrepancies between the datasets as illustrated in Fig. 9. Seasonal periods are defined as DJF (December–

January–February), MAM (March–April–May), JJA (June–July–August), and SON (September–October–November).

For the total ozone column, the scatterplots show an excellent agreement between IASI-CDR and TROPESS-CrIS data, with a bias consistently below 1.2% and correlation coefficients higher than 0.97 across all seasons. This indicates a very strong performance of both datasets in capturing total ozone variability. For the tropospheric ozone column, the differences are



more pronounced, with biases ranging from 10% to 12% and correlation coefficients varying between 0.77 during DJF
(when sensitivity is lower) and 0.91 during JJA (when sensitivity is larger), reflecting a relatively weaker, yet still
reasonable, agreement.

The spatial distribution analysis provides further insights into these discrepancies. For the total ozone column, CrIS
systematically underestimates ozone concentrations at northern high latitudes and in the southern midlatitudes, while
overestimating ozone in the tropics across all seasons. The differences are larger in DJF and lower in JJA, when IASI and
CrIS sensitivities are lower and higher, respectively. In the case of the tropospheric ozone column, CrIS tends to
underestimate ozone levels across most regions, with the notable exception of a prominent red band at the southern
midlatitudes, reflecting the overestimation pattern seen in the total ozone column. This specific feature may represent the
impact of persistent cloud cover on the CrIS sensitivity and therefore retrieval. The differences in tropospheric ozone
columns between IASI and CrIS are partly driven by the a priori information used in each retrieval, which is fixed for IASI
and variable (both in latitude and season) for CrIS. This is illustrated in Fig. B1 in Appendix B, which shows similar patterns
to the observed differences.

Overall, while the performance of both datasets is excellent for total ozone, the tropospheric ozone comparisons highlight
areas requiring further investigation, particularly regarding the systematic biases observed in the southern midlatitudes.

## 3.2 Comparison with ozonesonde data

Figure 10 (top panels) presents a comparison of the mean retrieved IASI, smoothed, and raw sonde profiles across three
latitude bands, representing the tropics, midlatitudes, and polar regions, based on all IASI-sonde coincidences from the 16-
year period (2008–2023). The top panels show that IASI effectively captures the main features of the ozonesonde vertical
distribution, including: (i) ozone peaks around 20–30 km, varying with latitude, (ii) a downward shift in the altitude of the
ozone maximum as latitude increases, and (iii) pronounced ozone gradients at the boundary between the troposphere and the
stratified lower stratosphere. Additionally, a secondary second smaller peak around 200 hPa appears in IASI data over polar
regions but is not consistently seen by ozonesondes. Likely linked to stratosphere-troposphere exchange, it may result from
polar vortex dynamics in winter and early spring. Since IASI data are averaged over a 1° area around the sonde station, it
captures greater spatial and temporal variability, possibly detecting stratosphere-troposphere exchange events that sondes,
with single-time profiles, may miss. Further analysis is needed to confirm its origin.
While IASI struggles to accurately represent the gradient toward the tropopause and the ozone peak, particularly in the
tropics, it successfully captures the peak amplitude and changes in vertical gradients. This issue is less pronounced in the
midlatitudes and polar regions. However, the variability across different latitude bands becomes evident when analyzing the
standard deviations. As noted, variability is significantly lower in the tropics compared to the mid and high latitudes. This
pattern aligns with the more stable atmospheric conditions in the tropics, where ozone distribution remains relatively
uniform. In contrast, the mid and high latitudes experience greater fluctuations due to more dynamic atmospheric processes,





such as stronger circulation patterns and seasonal changes. These factors contribute to higher variability in ozone profiles, especially in the higher latitudes.

Figure 10 (bottom panels) illustrates the differences in ozone concentration (in Dobson units, DU) between the IASI retrieved and sonde profiles, both raw and smoothed. IASI shows lower $O_3$ values in the troposphere and middle

stratosphere, while overestimating $O_3$ in the upper troposphere and lower stratosphere (UTLS) throughout all latitude regions. Potential factors contributing to the increased bias in the UTLS include the limited vertical resolution of IASI, spectroscopic uncertainties related to ozone absorption lines, or the use of inadequate a priori information (Boynard et al., 2016). Specifically, the effect of using a priori constraints that change with latitude still needs to be explored further. In the tropics, smoothing the sonde data reduces differences compared to the raw data, particularly in the stratosphere, and

introduces a vertical shift of a few kilometers downward. However, in the UTLS, the differences remain unchanged. At midlatitudes, the differences between IASI and sonde profiles are generally smaller than those observed in the tropics. Additionally, the differences are less pronounced when comparing IASI data to the smoothed sonde profiles, suggesting that smoothing helps to reduce some of the variability and improve agreement with IASI data in this region. In the polar regions, the most significant differences occur around the 200 hPa level, coinciding with the secondary peak observed in the IASI

data. Here, the differences are more pronounced when comparing IASI data with smoothed sonde profiles than with raw sonde profiles. This is likely because the reduced sensitivity of IASI at low temperatures can amplify discrepancies in the measurements. The figure highlights regional variations in the alignment between IASI and sonde data, emphasizing the impact of smoothing on data comparison. These differences underscore the importance of a priori assumptions in the retrieval process, especially in the tropics, where low ozone variability and unique atmospheric conditions make the retrieval

particularly sensitive to these assumptions. Further refinement of a priori constraints is necessary in the tropics to reduce biases, and smoothing is essential in improving agreement between IASI and sonde data, particularly in regions with more variable ozone profiles, such as the midlatitudes.

Figure 11, illustrating the scatterplots between IASI and smoothed sonde data, shows that IASI underestimates the tropospheric ozone column, with biases ranging from -2% in the tropics to -10% in the midlatitudes and polar regions (see

Fig. 8 for the spatial distribution of the bias). The standard deviation is about 11% across all latitude bands, indicating a consistent level of variability in the measurements. The correlation coefficient is high (0.9) in both the tropics and the midlatitudes, suggesting strong agreement between IASI and sonde concentrations, but drops to 0.6 in the high latitudes, indicating weaker performance in the polar regions, which is due to limited signal in those regions (DOFS=0.5). The DOFS are 1.5 in the tropics and 1 in the midlatitudes, reflecting better retrieval of information in those regions. The number of

observations is 2-3 times higher in the midlatitudes compared to the tropics and the high latitudes, which could contribute to the more robust results in those regions. The comparison between IASI and raw sonde data also included in Fig. 11 shows similar results to the comparison with smoothed sonde data, although it exhibits a lower bias in the tropics and a larger bias in the polar regions. The higher bias observed with smoothing in the polar regions may be due to the loss of important





temperature and humidity variability, while the lower bias in the tropics could result from the more stable atmospheric
conditions, which are better captured by smoothing.

## 4. Tropospheric O₃ trends

### 4.1 Regional trends

Figure 12 shows the spatial distribution of tropospheric ozone trends from satellite data for two periods: 2008-2019 (before
the COVID-19 pandemic) and 2008-2023 (including the pandemic). The results show distinct patterns across the two periods
analyzed. From 2008 to 2019, ozone concentrations decrease primarily in the tropics, with increases in certain regions, such
as over the tropical Pacific, in the Arabian Peninsula and in Asia. From 2008 to 2023, tropospheric ozone concentrations
exhibit a clear downward trend, particularly in the tropics and in Europe. However, the Arabian Peninsula and the North
China Plain (NCP) experience slight increases in ozone levels, though these are less pronounced compared to the 2008-2019
period. The impact of the 2020 lockdowns, which led to a reduction in ozone precursors ($NO_x$ and VOCs), likely contributed
to a decrease in ozone production. However, regional emissions and other factors may help explain the continued positive
ozone trends in the Arabian Peninsula and NCP. This finding in Asia agrees with Dunn et al. (2024) who found the strongest
positive trends above South and East Asia over 2004-2023, based on OMI-MLS observations.

When examining trends in the upper (450 hPa–tropopause) and lower (surface–450 hPa) troposphere for the periods 2008-
2019 and 2008-2023 (see Fig. C1 and C2 in Appendix C), negative trends with high certainty ($p \leq 0.05$) are consistently
observed in the lower troposphere for both periods. The upper troposphere presents a more complex picture: from 2008 to
2019, positive trends with high certainty are evident in several midlatitude regions, including the North Pacific Ocean, Asia,
and the Mediterranean, while negative trends with high certainty are observed in the tropical Pacific Ocean. From 2008 to
2023, most regions exhibit weaker ozone trends, except for the North Pacific Ocean, the Arabian Peninsula, and the NCP,
where positive trends with high certainty are observed. These findings align with Dufour et al. (2025), who reported negative
trends with high certainty in the lower troposphere and trends with low to medium certainty in the upper troposphere, except
over China, based on IASI-O3 KOPRA data from 2008 to 2022.

In the upper troposphere, the tropics generally exhibit persistent negative trends in ozone, except in areas affected by
recurrent fires which show a different behavior. In these regions, ozone levels can increase due to the release of large
amounts of CO and VOCs during fires, which trigger photochemical reactions that produce ozone, particularly in the
presence of sunlight. This phenomenon is especially pronounced in the upper troposphere, where ozone formation is more
sensitive to photochemical processes. The upward transport of pollutants from fires can lead to an accumulation of ozone in
the upper troposphere, contributing to the observed trends in these regions. Additionally, the long-range transport of ozone
precursors, particularly in the free troposphere, as documented by Glotfelty et al. (2014) and Itahashi et al. (2020), further
enhances these positive trends. At mid and high latitudes, the observed positive trends in the upper troposphere may reflect



stratospheric influences, as indicated by the limited DOFS in these regions (see Fig. 4), possibly due to transport processes, such as stratosphere-troposphere exchanges (Li et al., 2024). In contrast, the DOFS in the tropics is around 2, allowing for a clearer separation of the upper and lower troposphere without significant stratospheric contribution. Separating the surface-450 hPa and 450 hPa-tropopause columns provides valuable insights, as it reveals decreasing ozone trends in the tropics that are not observed at mid and high latitudes, except in regions impacted by recurrent fires.

In order to better understand the spatial distribution of the trends, Figure 13 shows the monthly time series of tropospheric ozone anomalies derived from the IASI-CDR product for different latitude bands and regions (see Fig. 12 for the definition of the regions). An interesting feature from this figure is the negative tropospheric ozone anomalies starting in 2020 across most of the regions (except for the Asia and the Nino 3.4 regions), continuing until 2023. This drop in 2020 is also observed in the OMI-MLS tropospheric ozone product and is attributed to the reduced emissions of ozone precursors across the NH
due to COVID-19 lockdown restrictions (Ziemke et al., 2022). These negative anomalies, beginning during the COVID-19 period, may partially explain the observed negative trends with medium certainty in the northern midlatitudes (p-value=0.05), high certainty in the southern mid-latitudes (p-value=0.02), and very high certainty in the tropics (p-value≤ 0.01) over the 2008–2023 period. Another factor, which could contribute to the persistence of the negative anomaly over several years is the three consecutive years of La Niña conditions in the tropical Pacific Ocean (from mid-2020 to early
2023), followed by a rapid transition to ENSO-neutral and then El Niño conditions by May (Dunn et al., 2024). The El Niño 3.4 index for 2008-2023 are also plotted on Fig. 13 (in gray). We can note an anti-correlation between La Niña events (negative anomalies) and ozone anomalies in the Nino 3.4 region, a pattern not observed in the other regions. For the period 2008–2019, trends with low to very low certainty are found, except in Asia (a positive trend of $+0.23\pm0.10\%$ year$^{-1}$; p-value=0.03; high certainty) and the tropics (a negative trend of $-0.20\pm0.10\%$ year$^{-1}$; p-value=0.04; high certainty). However,
caution is advised regarding the anomaly values at the start and end of the study period, as they appear to influence the trends (e.g. Pope et al., 2024; Gaudel et al., 2024).

Figure 14 shows the annual and seasonal trends in tropospheric ozone over the two periods of study (2008-2019 and 2008-2023) for different regions and three tropospheric columns: full (following the WMO thermal definition), lower (surface to 450 hPa) and troposphere (450 hPa to thermal tropopause) troposphere. Over the period 2008-2019, no seasonal trend in
tropospheric ozone is found for all regions, except in the tropics and in the Nino 3.4 region characterized by negative trends with high certainty in JJA/SON and in MAM, respectively. In the tropics, the annual trend in tropospheric ozone is driven by contributions from both the lower and upper troposphere in JJA, and solely by the lower troposphere in SON. In the Nino 3.4 region, the annual trend is primarily influenced by both the upper and lower troposphere in MAM. Over the period 2008-2023, trends in the upper troposphere are less positive, while those in the lower troposphere are more negative across all
regions. This results in more negative trends in the full tropospheric ozone column compared to 2008-2019. A negative trend with high certainty in the full tropospheric ozone column is found in MAM and SON in Europe and North America, respectively. Asia shows no annual and seasonal trend for the full tropospheric column over 2008-2023 while exhibiting positive annual trends with high certainty during 2008-2019, primarily due to positive trends in the upper troposphere in



MAM and SON. For both periods, trends with low to very low certainty in the full tropospheric column are generally due to

compensating positive and negative trends in the upper and lower troposphere, respectively.

To assess the tropospheric ozone trends derived from IASI data, we compare them with trends calculated from sonde data. As described in Section 2.3, only ozonesonde stations with at least three observations per month (to compute monthly means) and a minimum of 70% data coverage over the period of study are included in the trend analysis. Over the 43 stations considered, seven meet these criteria, including four in Europe (Uccle, Payerne, Hohenpeissenberg, and Madrid), one in

North America (Boulder), one in the tropics (Hilo), and one in the southern midlatitudes (Lauder). Due to the geographic distribution of the selected stations, the global trends assessed in this study are primarily representative of Europe.

Figure 15 (left panels) illustrates the monthly time series of tropospheric ozone anomalies derived from IASI and smoothed sonde data, for the global scale and Europe during the 2008–2023 period. It shows a strong agreement in the temporal variability of tropospheric ozone anomalies between IASI and smoothed sonde data on both scales. A pronounced drop in

tropospheric ozone is observed in 2020, coinciding with the COVID-19 pandemic. Negative trends in tropospheric ozone with very high certainty are identified, with slightly higher magnitudes derived from IASI (-0.40 ±0.10% $yr^{-1}$; p-value=0.00) compared to smoothed ozonesonde data (-0.27±0.10% $yr^{-1}$; p-value=0.01) on the global scale. Similar findings are obtained with raw sonde data.

Figure 15 (right panels) show the corresponding monthly biases between IASI and ozonesonde data. At the global scale, a

weak drift with medium certainty is found between IASI and smoothed sonde data (-0.08 ±0.04% $yr^{-1}$; p-value=0.08). However, in Europe, a negative drift with high certainty (-0.12±0.05% $yr^{-1}$; p-value=0.02) is observed, suggesting a potential instability in either the IASI or sonde measurements. To further investigate the source of the negative drift between IASI and sonde data in Europe, we analyzed the seven individual stations, which is presented in the next section.

## 4.2 Local trends

Figure 16, illustrating the monthly time series of tropospheric ozone anomalies (left panels) and differences between IASI and sonde (right panels) for individual stations, indicates that the observed negative drift in Europe (see Fig. 15; right) is primarily driven by the drift with very high certainty observed at the Payerne station (-0.55±0.09% $yr^{-1}$; p-value=0.00). The observed drift at Payerne seems to be associated with an underestimation of tropospheric ozone anomalies by IASI at this location, but the exact cause remains unclear, as it leads in larger negative trends in IASI data compared to the sonde

measurements. After excluding the Payerne station and recalculating the trends and drifts at both the global scale and in Europe, a negative trend in tropospheric ozone with high certainty remains, but no drift is observed. Further investigation is required to understand the drift observed in Payerne.

For most of individual stations, IASI and sonde data show trends in tropospheric ozone with medium to low certainty. However, at Uccle both IASI and sonde show clear negative trends in the range 0.53-0.79% $yr^{-1}$ with high certainty (p-




value≤0.02). At Hohenpeissenberg, IASI data show negative trends with high certainty (p-value = 0.03), while sonde data indicates negative trends with moderate certainty (p-value = 0.06).

## 5. Conclusions

This study, conducted within the framework of the TOAR-II project, aims to deepen our knowledge of the spatiotemporal patterns and trends in global tropospheric ozone, using the $O_3$ Climate Data Record (CDR) from the IASI/Metop (IASI-CDR). This dataset, recently reprocessed by EUMETSAT AC SAF, provides a consistent 16-year record (2008–2023) for the first time, enabling robust ozone trend analysis and contributing to our understanding of tropospheric ozone variability. The research focused on two main objectives: evaluating the quality and consistency of the IASI-CDR $O_3$ products and investigating the spatiotemporal variability and long-term trends in tropospheric ozone.

Key findings are summarized as follows:

**Assessment of IASI-CDR ozone data:**

- A comparison with CrIS-TROPESS data shows excellent agreement for total ozone (biases < 1.2%, correlations > 0.97) and good agreement for tropospheric ozone (biases 10–12%, correlations 0.77–0.91). However, the systematic overestimations in CrIS tropospheric ozone, particularly at high northern and southern midlatitudes, indicate the need for further investigation.

- When compared with ozonesonde data, IASI-CDR profiles generally capture ozone distribution features well, but tend to underestimate tropospheric and stratospheric ozone, and overestimate UTLS ozone. The IASI-CDR product underestimates the tropospheric ozone column, with biases ranging from -2% in the tropics to -10% at mid and high latitudes.

- A regional drift is observed in the IASI-sonde difference in Europe (-0.12 ± 0.05%/yr), primarily associated with the Payerne station. On a global scale, no drift is found, and excluding Payerne, no drift is observed in Europe. Further investigation is needed to ensure consistency in long-term ozone trend assessments.

**Spatiotemporal ozone variability and trends:**

- Tropospheric ozone trends derived from IASI data reveal regional variability. From 2008 to 2019, ozone decreased in the tropics (-0.20% year$^{-1}$), but increased in localized regions like North Pacific Ocean, Asia and the Arabian Peninsula. Over the extended period (2008–2023), negative trends became more pronounced, particularly in the tropics and in Europe, while positive trends weakened.

- Seasonal and regional analyses suggest that surface emissions, atmospheric transport, and events such as the 2020 COVID-19 lockdowns significantly influenced ozone dynamics. Negative trends are more pronounced in the lower troposphere and the tropical upper troposphere. In contrast, positive trends in the upper troposphere are generally observed with low certainty, except for localized increases associated with fire emissions and transport processes.



- A comparison of IASI and sonde trends from seven well-covered stations during 2008-2023 showed strong consistency in variability, with global trends of $-0.40 \pm 0.10\%$ year$^{-1}$ for IASI and $-0.27 \pm 0.10\%$ year$^{-1}$ for sonde data, both with very high certainty (p-value≤0.01). Both datasets also observed a drop in ozone in 2020, attributed to the pandemic.

**Comparison with UV datasets:**

IASI-CDR data reveal negative trends in tropospheric ozone for the period 2008-2023, contrasting with the positive or stable trends observed in UV satellite data (e.g., OMI, 2004-2019) and in situ measurements (1990-2019 and 1998-2023) (Gaudel et al., 2024; Elshorbany et al., 2024; Thompson et al., 2024). This discrepancy may arise from several factors:

- Temporal coverage: IASI captures trends starting in 2008, a period marked by stricter emission controls in
industrialized regions, likely reducing ozone precursors such as $NO_x$ and VOCs. In contrast, OMI and in situ data span earlier periods when emissions in emerging economies were rising, potentially masking recent declines. This discrepancy highlights the importance of aligning the analysis period across observational datasets to ensure a meaningful comparison of trends.

- Vertical sensitivity differences: IASI is more sensitive to the middle and lower troposphere, whereas OMI primarily
detects ozone in the upper troposphere and lower stratosphere. This difference in sensitivity could help explain the contrasting trends in tropospheric ozone observed. In support of this, Pimlott et al. (2024) also identified negative trends in lower tropospheric ozone from 2008 to 2017, based on UV satellite data.

- A priori constraints: Both IASI and OMI data rely on a priori constraints, which represent another factor that can influence trend results. If these constraints are misaligned with actual ozone profiles, it can lead to discrepancies,
especially in regions like the tropics where ozone levels in the UTLS are naturally low.

- Other source of differences: Differences in atmospheric dynamics, regional variability, and calibration or sampling methods further contribute to the observed trends. A more detailed intercomparison of these datasets, along with ground-based measurements, is essential to resolve these differences.

**Implications and future directions:**

This study demonstrates the value of the IASI-CDR product as a reliable resource for long-term tropospheric ozone monitoring and trend analysis. The findings offer new insights into tropospheric ozone variability and highlight the need for improved a priori constraints, especially in the tropics and UTLS regions, to enhance retrieval accuracy and reduce biases. This is supported by Keppens et al. (2025), who also utilized the IASI-CDR tropospheric ozone product and emphasized the importance of harmonizing satellite ozone measurements to improve consistency and reliability, despite challenges related to
vertical smoothing and measurement uncertainty.

The period of tropospheric ozone decline from 2020 to 2023 suggests a potential shift in ozone trend trajectories, emphasizing the importance of continued monitoring and a deeper understanding of the processes governing tropospheric ozone dynamics. Extending the analysis through 2023 has revealed a notable convergence: both UV and IR satellite




instruments now capture a similar feature, specifically the pronounced drop in tropospheric ozone starting in 2020. This consistency underscores the importance of long-term, homogeneous datasets for reliable trend analysis.

## 6. Appendices

### Appendix A: Validation of IASI ozone subcolumns

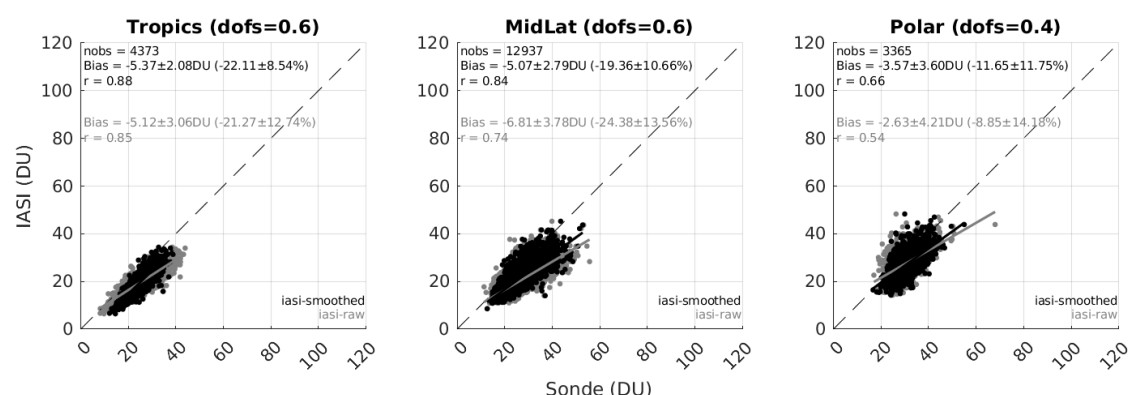


**Figure A1: Comparison of IASI with raw (gray) and smoothed (black) sonde surface – 300 hPa ozone subcolumns for the period 2008-2023, shown across three latitude bands: tropics (30°S-30°N), midlatitudes (30°-60° North and South) and polar regions (60°-90° North and South). The 1:1 line (dashed) and the linear regression lines (black for smoothed, gray for raw) are shown on each scatterplot, along with statistics including the number of collocations, mean bias with standard deviation in both Dobson units (DU) and percent (%), correlation coefficient (r), and the associated mean Degrees of Freedom for Signal (DOFS).**

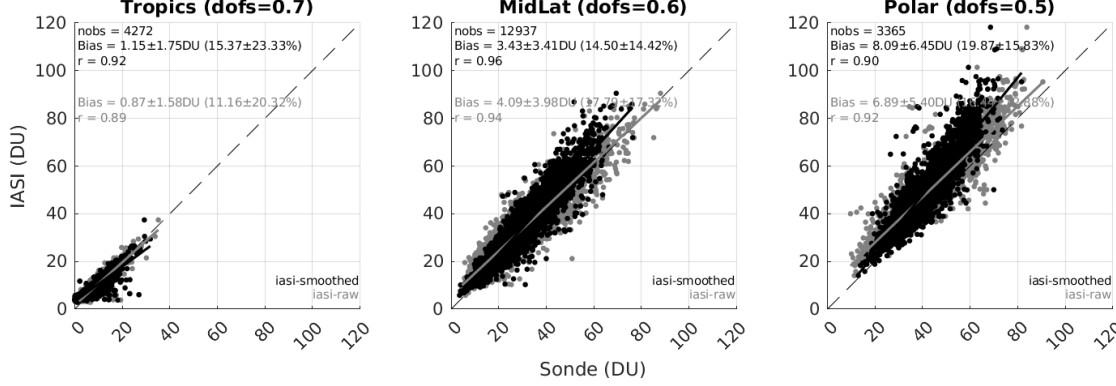

**Figure A2: Same as Fig. A1, but for the 300 – 150 hPa subcolumn.**





## 150-25 hPa ozone subcolumn

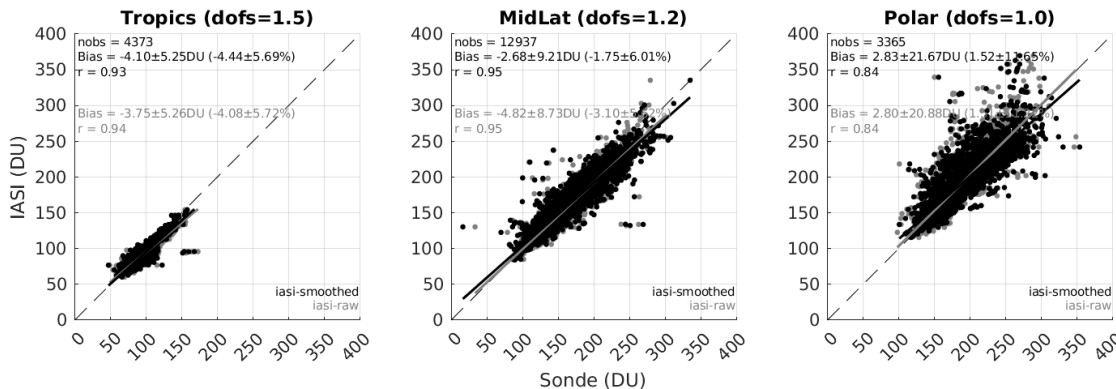


**Figure A3: Same as Fig. A1, but for the 150 – 25 hPa subcolumn.**

## 25-3 hPa ozone subcolumn

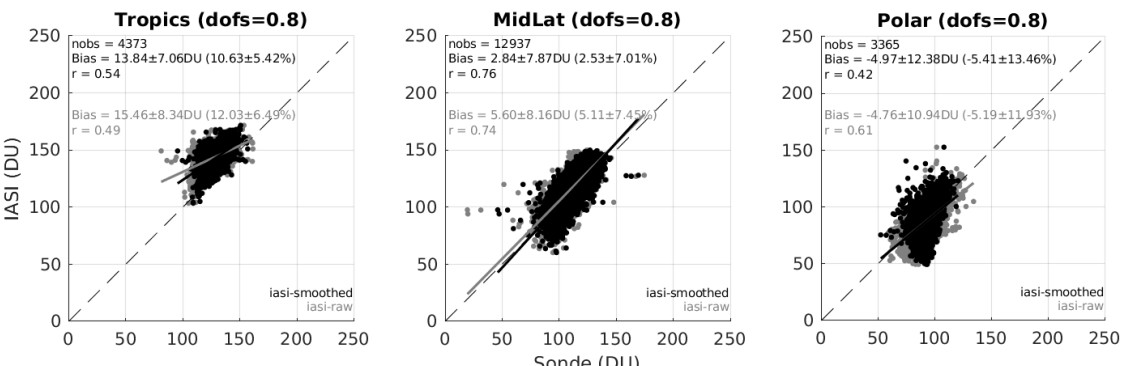

**Figure A4: Same as Fig. A1, but for the 25– 3 hPa subcolumn.**




**Appendix B: IASI and CrIS a priori data**

**Figure B1: Spatial and seasonal distribution of IASI and CrIS a priori surface-300 hPa ozone column over the period 2016-2022 in Dobson units (DU). The data are averaged on a global grid with a resolution of 1°x1°. The a priori profile for IASI is constant, so there is no seasonal variation, while the a priori data for CrIS varies across seasons.**




**Appendix C: Lower and upper tropospheric ozone trends derived from IASI-CDR**

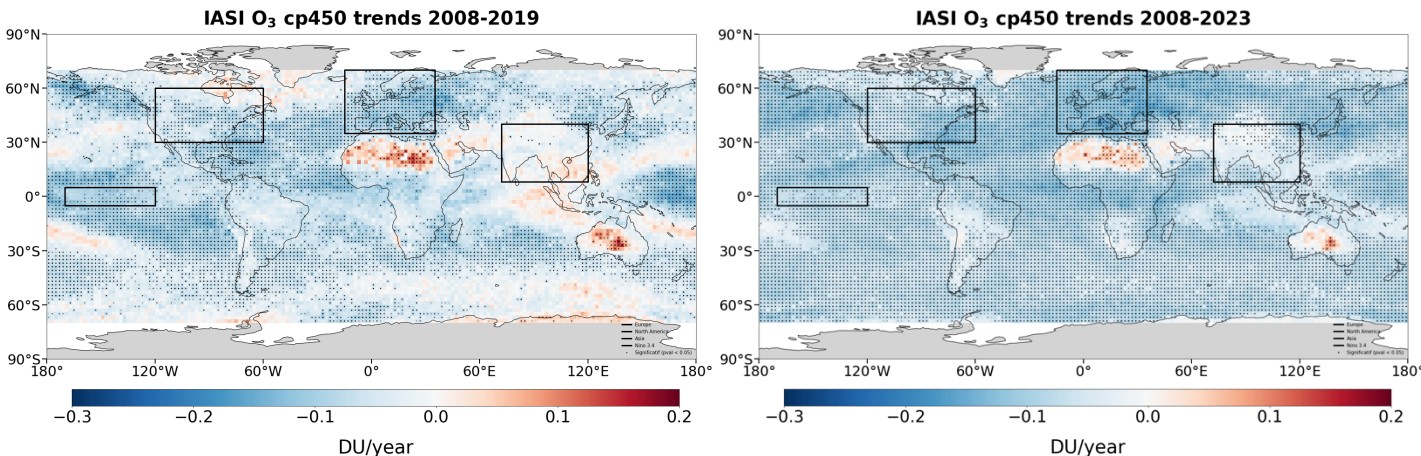

**Figure C1: Spatial distribution of lower tropospheric ozone (surface-450 hPa) trends (in DU year$^{-1}$) derived from IASI-CDR before the COVID-19 pandemic (2008–2019) and including the pandemic (2008–2023). Dots indicate trends with 95% confidence. The black rectangles indicate the boundaries of the regions analyzed in Fig. 13.**


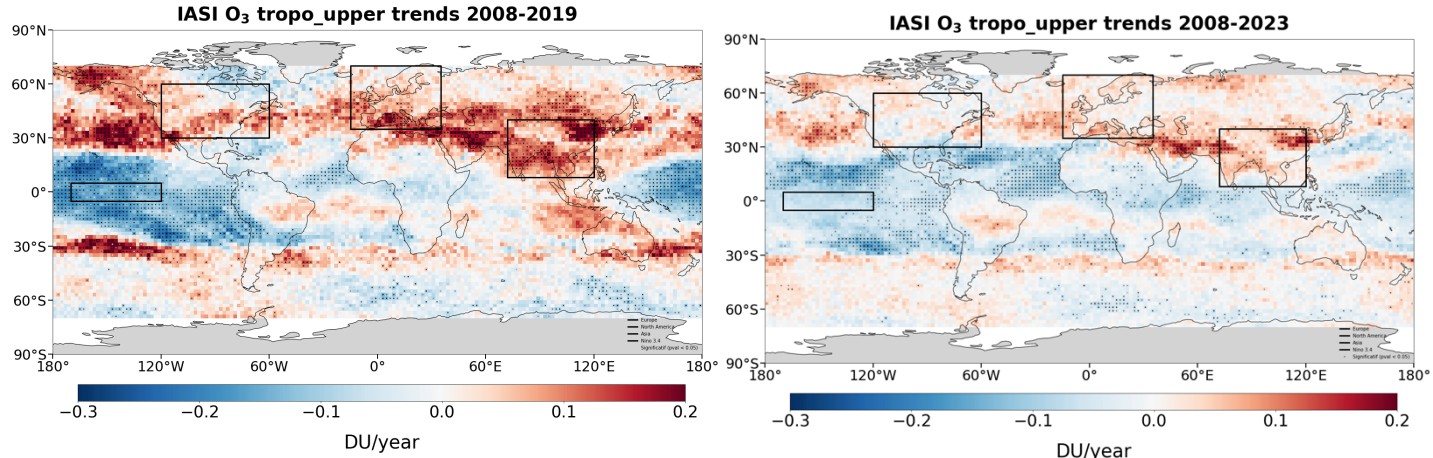

**Figure C2: Spatial distribution of upper tropospheric ozone (450 hPa-tropopause) trends (in DU per year) derived from IASI-CDR before the COVID-19 pandemic (2008–2019) and including the pandemic (2008–2023). The tropopause altitude is estimated using the WMO thermal definition. Dots indicate trends with 95% confidence. The black rectangles indicate the boundaries of the regions analyzed in Fig. 13.**




**Data availability**

The IASI-CDR $O_3$ data can be downloaded from the Aeris portal (http://iasi. aeris-data.fr/O3) and includes daily files generated from AC-SAF O3-CDR orbit files. The TROPESS-CrIS $O_3$ data are available at the NASA GES-DISC database (10.5067/8TS8WCSCZJMV, TROPESS-CrIS, 2023). The ozonesonde data can be downloaded from TOAR-II Harmonization and Evaluation of Ground-based Instruments for Free-Tropospheric Ozone Measurements (HEGIFTOM) working group (https://hegiftom.meteo.be, HEGIFTOM, 2025), from the World Ozone and Ultraviolet Radiation Data Centre (WOUDC; https://doi.org/10.14287/10000008; WMO/GAW Ozone Monitoring Community, 2025), from NOAA-ESRL database (http://www.esrl.noaa.gov/gmd/dv/ftpdata.html; NOAA, 2025), and from the Southern Hemisphere Additional Ozonesondes ( https://tropo.gsfc.nasa.gov/shadoz/, SHADOZ, 2025).

**Author contribution**

AB designed the study, performed data analysis and led the writing of the paper. All co-authors provided data and/or contributed to the discussion and improvement of the paper.

**Competing interests**

The authors declare that they have no conflict of interest.

**Acknowledgments**

IASI is a joint mission of EUMETSAT and the Centre National d'Etudes Spatiales (CNES, France). The IASI Level 1C data are distributed in near real time by EUMETSAT through the EUMETCast system distribution. The authors acknowledge the Aeris data infrastructure (https://www.aeris-data.fr/) for providing access to the IASI Level 1C data and Level 2 temperature data used in this study. This work was undertaken in the framework of the EUMETSAT AC SAF project (http://acsaf.org), the European Space Agency Ozone Climate Change Initiative (Ozone_cci, http://www.esa-ozone-cci.org) and the Copernicus Climate Change (C3S, https://cds.climate.copernicus.eu/). The French scientists are grateful to CNES and Centre National de la Recherche Scientifique (CNRS) for financial support.

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



**Tables**

**Table 1: IASI/Metop-B retrieval errors, including smoothing and the measurement error, along with the DOFS for total and tropospheric columns, averaged on the 15th of each month for both daytime and nighttime across 30° latitude bands in 2023.**

|  |  | Total column | | Tropospheric column | |
| --- | --- | --- | --- | --- | --- |
|  |  | Day | Night | Day | Night |
| Error (%) | 60°-90°N | 1.37±0.49 | 0.95±0.41 | 12.01±3.38 | 16.00±3.41 |
|  | 30°-60°N | 0.95±0.41 | 1.24±0.56 | 8.38±3.94 | 9.99±3.87 |
|  | 0°-30°N | 0.90±0.13 | 1.00±0.19 | 7.65±2.72 | 8.09±2.65 |
|  | 0°-30°S | 0.91±0 | 0.99±0.18 | 8.15±2.96 | 8.55±2.76 |
|  | 30°-60°S | 0.97±0.20 | 1.05±0.22 | 10.66±3.86 | 11.31±3.74 |
|  | 60°-90°S | 4.43±2.19 | 3.64±2.15 | 19.91±4.58 | 17.30±4.83 |
| DOFS | 60°-90°N | 2.85±0.37 | 2.54±0.32 | 0.52±0.30 | 0.31±0.23 |
|  | 30°-60°N | 3.29±0.32 | 3.15±0.33 | 1.06±0.37 | 0.95±0.38 |
|  | 0°-30°N | 3.71±0.16 | 3.68±0.17 | 1.70±0.17 | 1.68±0.17 |
|  | 0°-30°S | 3.71±0.17 | 3.67±0.17 | 1.69±0.17 | 1.67±0.17 |
|  | 30°-60°S | 3.30±0.22 | 3.27±0.21 | 1.07±0.33 | 1.02±0.31 |
|  | 60°-90°S | 2.65±0.33 | 2.54±0.34 | 0.32±0.22 | 0.43±0.25 |

**Table 2: List of the 43 sounding stations used for the validation exercise, including their name, geographical**
**coordinates (latitude and longitude in degrees), period of observation, averaged number of measurements per month, and sources. The data used in this study is sourced from HEGIFTOM (Harmonization and Evaluation of Ground-based Instruments for Free-Tropospheric Ozone Measurements). Incomplete HEGIFTOM time series are supplemented with data from WOUDC. Additional data from the NOAA and SHADOZ databases are also included. Stations highlighted in bold are used for trend analysis.**

| Station name | Latitude (°N) | Longitude (°E) | Period | averaged number of measurements per month | Sources |
| --- | --- | --- | --- | --- | --- |
| Alert | 82.50 | -62.34 | January 2008- June 2023 | 3.18 | HEGIFTOM/WOUDC |
| Eureka | 80.05 | -86.42 | January 2008- June 2023 | 4.51 | HEGIFTOM/WOUDC |
| NyAlesund | 78.92 | 11.92 | January 2008- July 2023 | 6.88 | HEGIFTOM/WOUDC |
| Resolute | 74.72 | -94.98 | January 2008- May 2023 | 2.27 | HEGIFTOM/WOUDC |
| Scoresbysund | 70.48 | -21.95 | January 2008- December 2023 | 3.61 | HEGIFTOM/WOUDC |
| Sodankyla | 67.36 | 26.63 | January 2008- December 2023 | 4.62 | HEGIFTOM/WOUDC |
| Lerwick | 60.13 | -1.18 | January 2008- September 2023 | 4.01 | HEGIFTOM/WOUDC |
| Churchill | 58.74 | -93.82 | January 2008- June 2023 | 2.67 | HEGIFTOM/WOUDC |
| Edmonton | 53.55 | -114.10 | January 2008- June 2023 | 3.38 | HEGIFTOM/WOUDC |
| GooseBay | 53.29 | -60.39 | January 2008- December 2022 | 3.41 | HEGIFTOM/WOUDC |





| | | | | | |
|---|---|---|---|---|---|
| Legionowo | 52.40 | 20.97 | January 2008- December 2023 | 3.70 | HEGIFTOM/WOUDC |
| Lindenberg | 52.22 | 14.12 | January 2008- October 2023 | 4.83 | HEGIFTOM/WOUDC |
| DeBilt | 52.10 | 5.18 | January 2008- December 2023 | 4.18 | HEGIFTOM/WOUDC |
| Valentia | 51.94 | -10.25 | January 2008- December 2023 | 2.27 | HEGIFTOM/WOUDC |
| **Uccle** | **50.80** | **4.36** | **January 2008- December 2023** | **9.89** | **HEGIFTOM/WOUDC** |
| Prague | 50.01 | 14.45 | January 2008- April 2023 | 3.36 | HEGIFTOM/WOUDC |
| KelownaPH | 49.94 | -119.40 | January 2008- June 2023 | 3.62 | HEGIFTOM/WOUDC |
| **Hohenpei** | **47.80** | **11.01** | **January 2008- December 2023** | **9.21** | **HEGIFTOM/WOUDC** |
| **Payerne** | **46.81** | **6.94** | **January 2008- December 2023** | **10.35** | **HEGIFTOM/WOUDC** |
| OHP | 43.92 | 5.71 | January 2008- November 2023 | 3.15 | HEGIFTOM/WOUDC |
| Yarmouth | 43.87 | -66.10 | January 2008- June 2022 | 3.55 | HEGIFTOM/WOUDC |
| TrinidadHead | 41.05 | -124.15 | January 2008- December 2023 | 4.07 | NOAA |
| **Madrid** | **40.45** | **-3.72** | **January 2008- December 2023** | **3.58** | **HEGIFTOM/WOUDC** |
| **Boulder** | **39.99** | **-105.26** | **January 2008- December 2023** | **2.90** | **NOAA** |
| Wallops | 37.90 | -75.70 | January 2008- November 2020 | 4.58 | HEGIFTOM/WOUDC |
| Tateno | 36.05 | 140.13 | January 2008- December 2023 | 3.03 | HEGIFTOM/WOUDC |
| Izana | 28.41 | -16.53 | January 2008- June 2023 | 4.27 | HEGIFTOM/WOUDC |
| Hongkong | 22.31 | 114.17 | January 2008- December 2023 | 3.43 | HEGIFTOM/WOUDC |
| Hanoi | 21.02 | 105.80 | January 2008- November 2021 | 1.48 | SHADOZ |
| **Hilo** | **19.72** | **-155.07** | **January 2008- December 2023** | **3.66** | **SHADOZ** |
| CostaRica | 10.00 | -84.11 | January 2008 - December 2023 | 3.03 | SHADOZ |
| Paramaribo | 5.81 | -55.21 | January 2008- December 2023 | 2.95 | SHADOZ |
| KualaLumpur | 2.73 | 101.70 | January 2008- December 2022 | 1.43 | SHADOZ |
| Nairobi | -1.30 | 36.80 | January 2008- June 2022 | 3.22 | SHADOZ |
| Natal | -5.40 | -35.40 | January 2008- August 2023 | 2.35 | SHADOZ |
| Ascension | -7.56 | -14.22 | January 2008- December 2023 | 2.62 | SHADOZ |
| Samoa | -14.25 | -170.56 | January 2008- October 2023 | 2.38 | SHADOZ |
| Fiji | -18.13 | 178.32 | January 2008- August 2023 | 1.50 | SHADOZ |
| Reunion | -21.10 | 55.50 | January 2008- December 2020 | 2.56 | SHADOZ |
| Irene | -25.90 | 28.20 | January 2008- March 2023 | 1.36 | SHADOZ |
| Broadmeadows | -37.69 | 144.95 | January 2008- December 2023 | 3.19 | HEGIFTOM/WOUDC |
| **Lauder** | **-45.04** | **169.68** | **January 2008- December 2023** | **4.71** | **HEGIFTOM/WOUDC** |
| Macquarie | -54.50 | 158.94 | January 2008- December 2023 | 3.36 | HEGIFTOM/WOUDC |




**Figures**

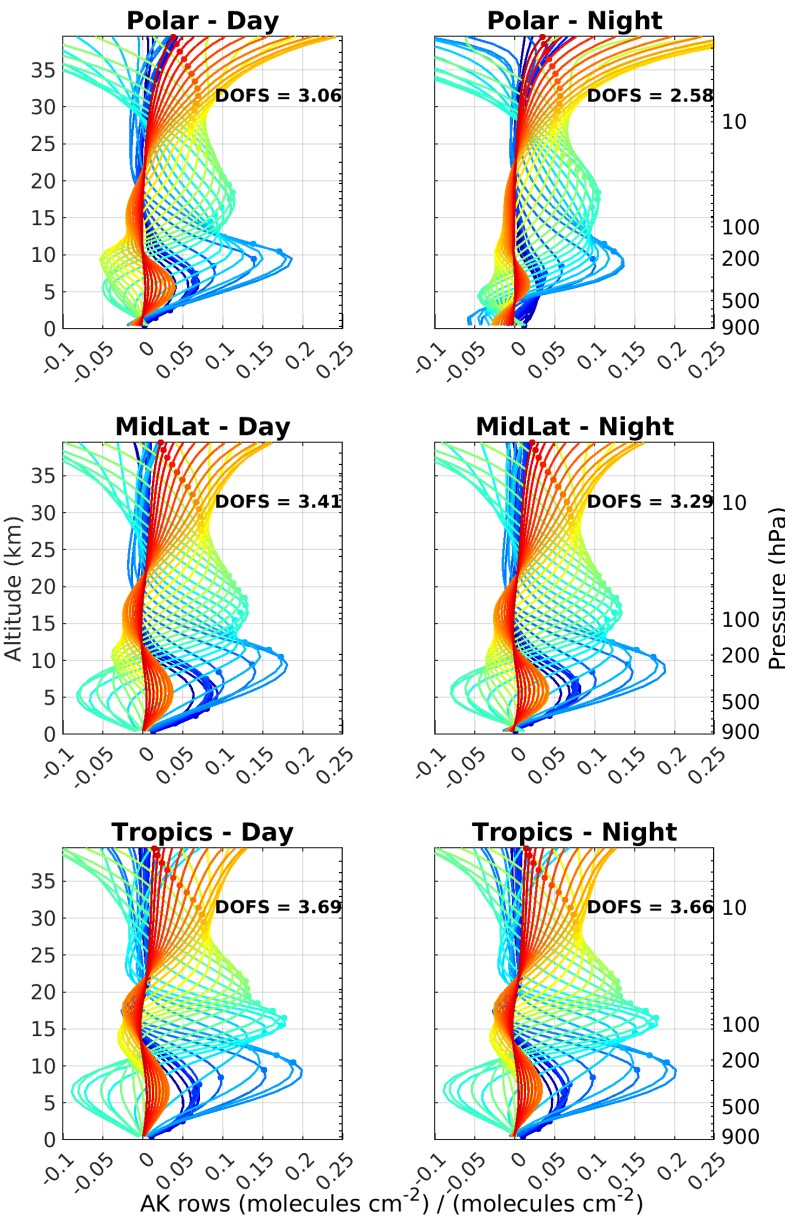

**Figure 1: Illustration of IASI-B averaging kernels (AK) averaged across three latitude bands (polar regions, midlatitudes and tropics) for 7 July 2023, showing the distribution of information in the vertical ozone profile. Each line represents a row of the AK matrix, and the circles indicate the altitude of each kernel. The polar regions, midlatitudes and tropics correspond to the following bands: 60°-90° North and South, 30°-60° North and South, and 30°S-30°N, respectively.**



**Figure 2: Spatial and seasonal distribution of total ozone column derived from IASI-CDR over the period 2008-2023 in Dobson units (DU): mean (left) and standard deviation (right). The data are averaged onto a global 1°x1° grid.**



**Figure 3: Monthly time series of IASI total ozone column derived from the CDR products in Dobson units (DU) for**
**Metop-A and Metop-B (in black and gray) over six 30° latitude bands. The time series of IASI-FORLI v20151001 product (Boynard et al., 2018) is also shown for the three Metop (in red, magenta and cyan). Note that the y-scale differs between the latitude bands. The vertical lines represent the date of the EUMETSAT L2 Product Processing Facility version changes (see Bouillon et al. (2020) for further details on these changes).**



**Figure 4: Same as Fig. 2, but for the tropospheric ozone column. The DOFS associated with the tropospheric column is also shown (right panels). The tropospheric column is integrated from the surface to the thermal tropopause, as defined by the World Meteorological Organization (WMO, 1957).**





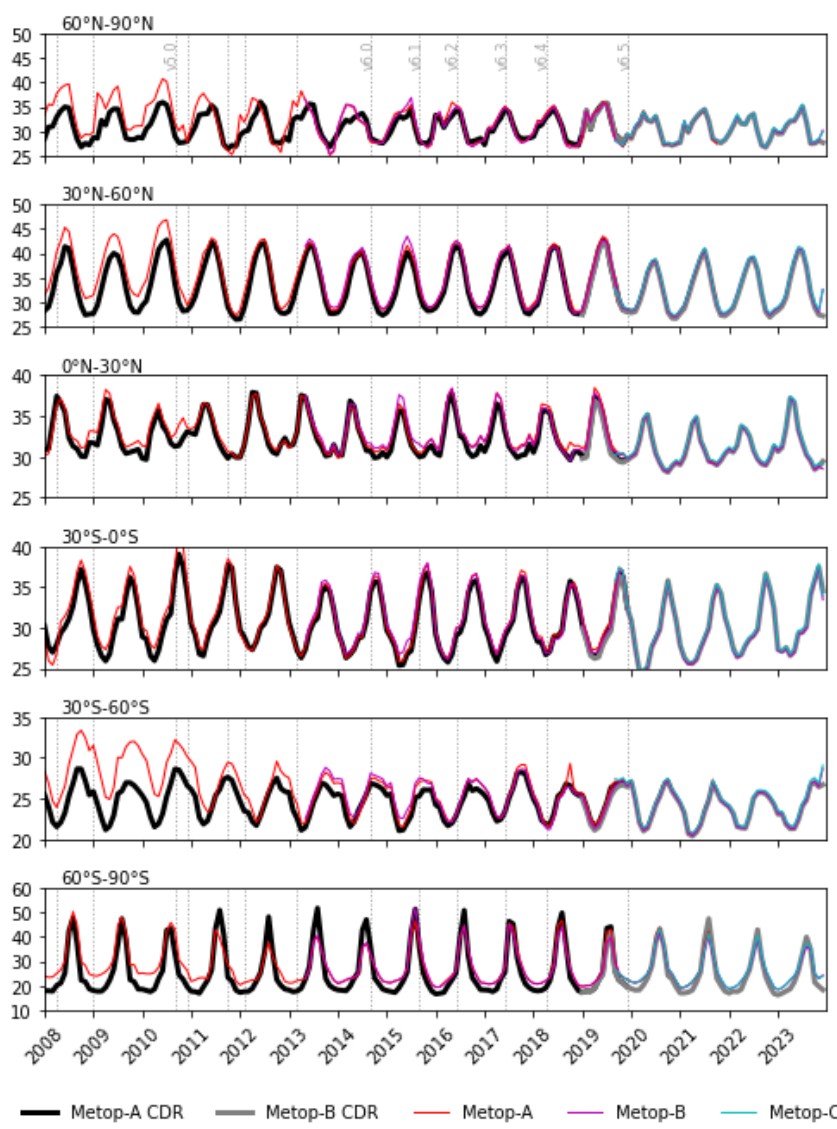

**Figure 5: Same as Fig. 3, but for the tropospheric column. The tropospheric column is integrated from the surface to the thermal tropopause, as defined by the World Meteorological Organization (WMO, 1957).**





**Figure 6: Spatial and seasonal distribution of total ozone column derived from CrIS data over the period 2016-2022 in Dobson units (DU): mean (left) and standard deviation (right). The data are averaged onto a global 1°x1° grid.**






**Figure 7: Same as Fig. 6, but for the tropospheric ozone column.**



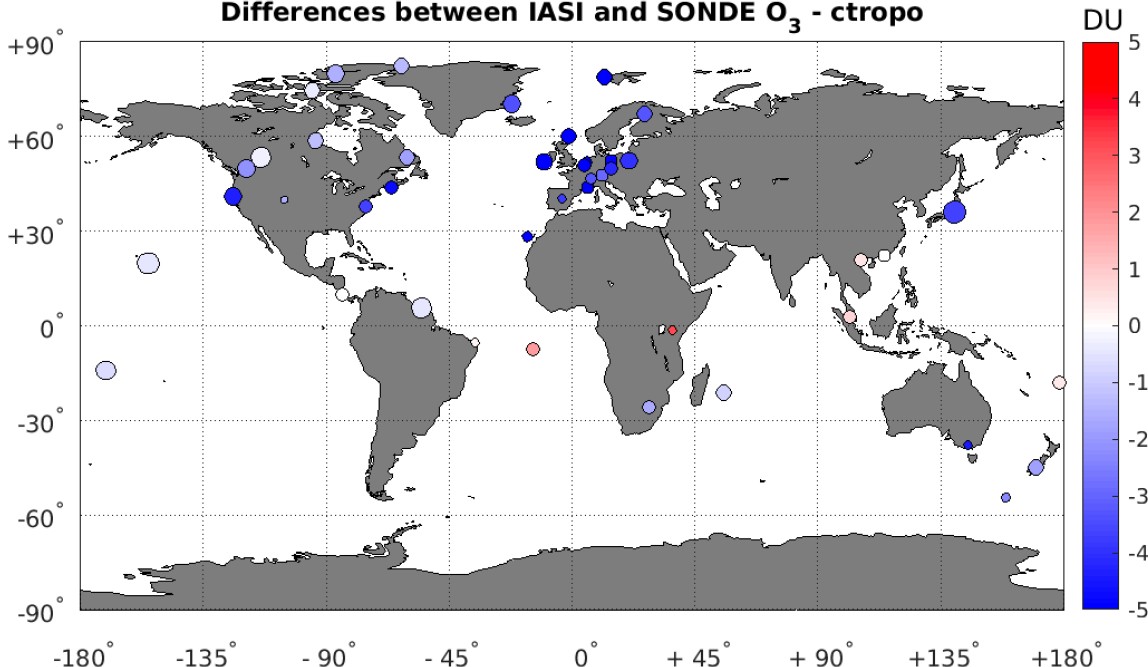

**Figure 8: Location of the 43 ozonesonde stations used in this study. The colors represent the mean difference between IASI and sonde tropospheric ozone column in Dobson units (DU) over the period from January 2008 and December 2023. The standard deviation is represented by the size of the circle.**




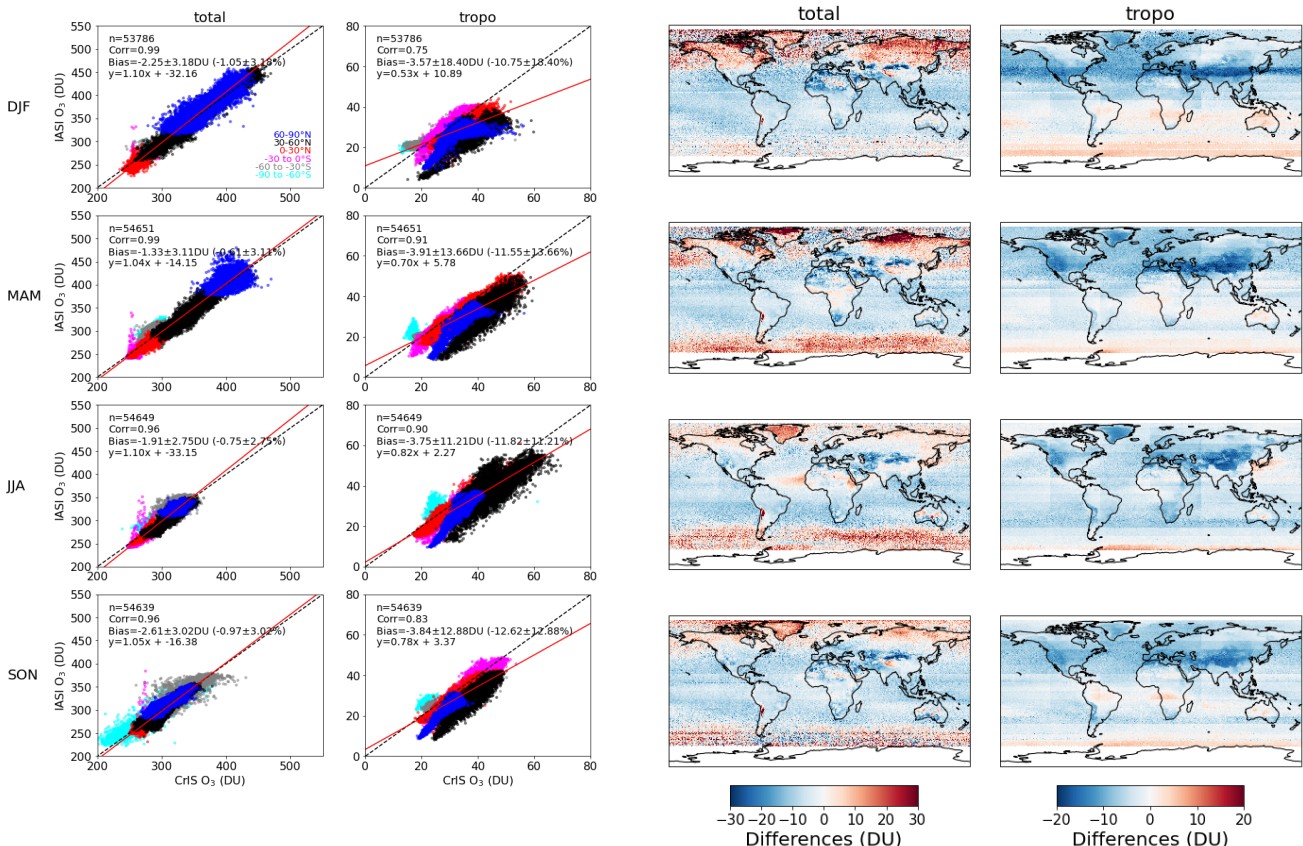

**Figure 9: Left columns: Seasonal scatterplots of IASI-CDR against TROPESS-CrIS total and tropospheric ozone columns for the period 2016-2022. The color code represents different latitude bands: blue (60°-90° N), black (30°-60° N), red (0°-30° N), magenta (0°-30°S), gray (30°-60° S), and cyan (60°-90° S). The 1:1 line (dashed black) and the regression line (red) are shown on each scatterplot, along with statistics including the number of common grid cells, the correlation coefficient, the mean bias with standard deviation in both Dobson units (DU) and percent (%), and the linear regression. The absolute and relative biases are calculated as: IASI-CrIS and [100 x (IASI-CrIS) / (0.5 x (IASI+CrIS))], respectively. Right columns: Spatial distributions of the differences between IASI and CrIS (in DU). DJF, MAM, JJA and SON represent December–January–February, March–April–May, June–July–August and September–October–November, respectively. The data are averaged onto a global 1°x1° grid.**



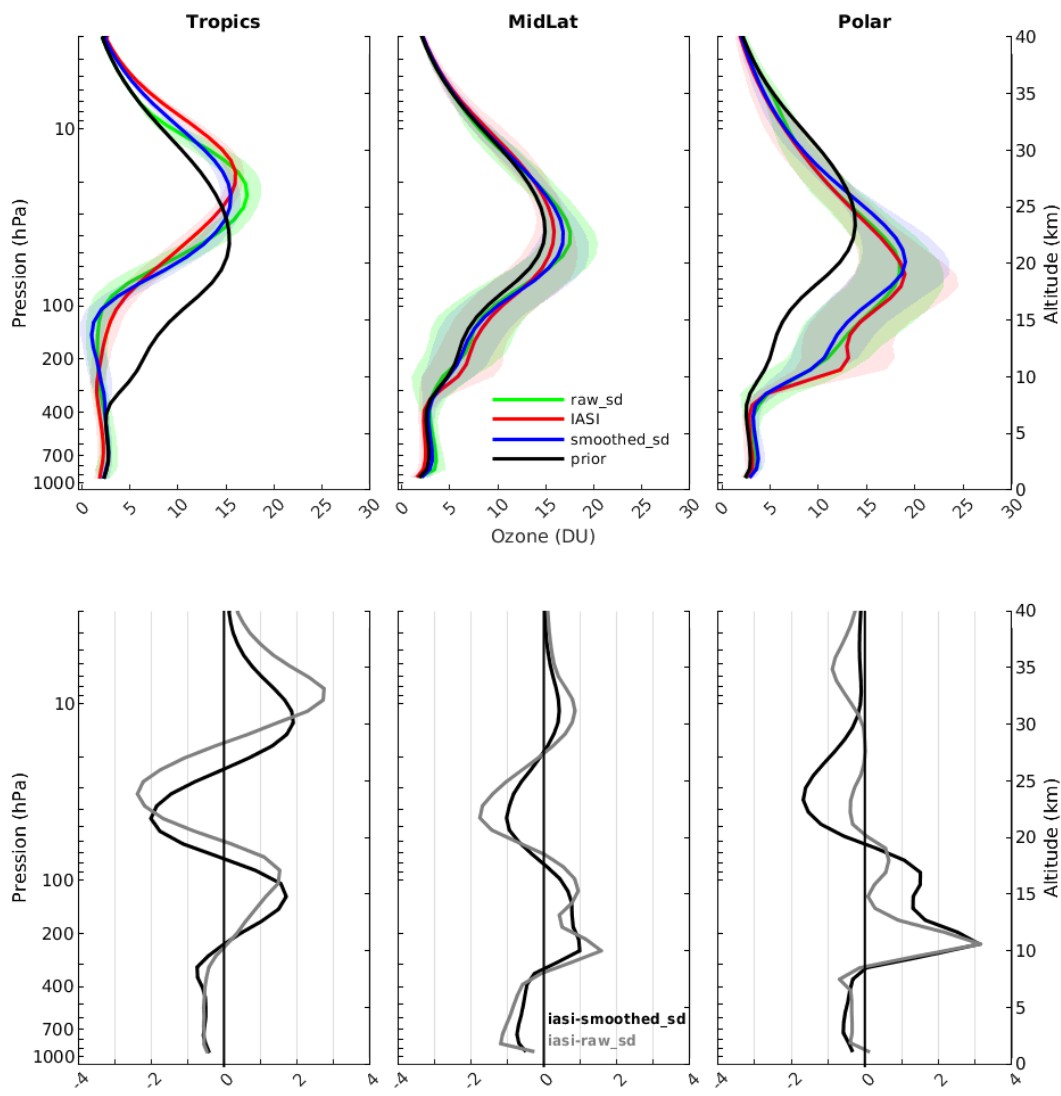

**Figure 10: Top panels: Mean ozone vertical profiles in Dobson units (DU) retrieved by IASI (red), measured by the sondes (green), and measured by the sondes after applying the IASI averaging kernels (blue) for three latitude bands: tropics (30°S-30°N), midlatitudes (30°-60° North and South) and polar regions (60°-90° North and South). The black line indicates the a priori ozone profile used in the IASI retrievals; Bottom panels: Vertical distribution of the differences in DU between the IASI retrieved mean profile and the smoothed sonde mean profile (black), as well as between the IASI retrieved mean profile and the raw sonde mean profile (gray).**





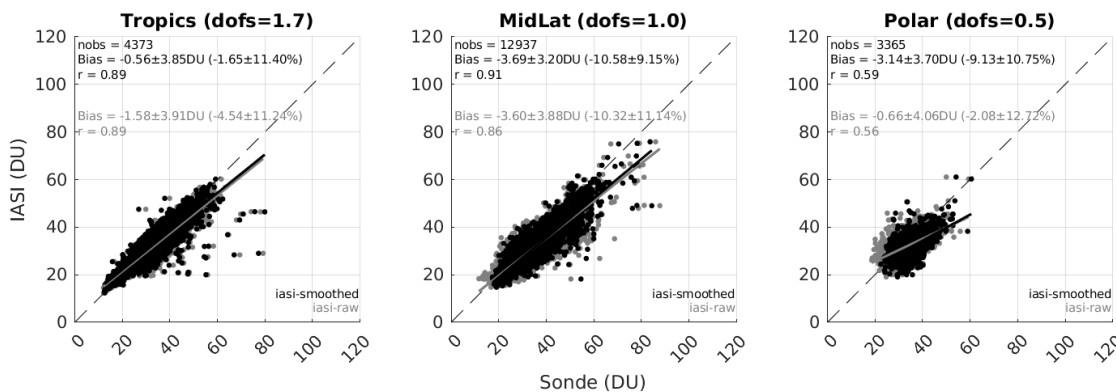

**Figure 11: Comparison of IASI with raw (gray) and smoothed (black) sonde tropospheric ozone subcolumns for the period 2008-2023, shown across three latitude bands: tropics (30°S-30°N), midlatitudes (30°-60° North and South) and polar regions (60°-90° North and South). The 1:1 line (dashed) and the regression lines (black for smoothed, gray for raw) are shown on each scatterplot, along with statistics including the number of collocations, mean bias with standard deviation in both Dobson units (DU) and percent (%), correlation coefficient (r), and the associated mean Degrees of Freedom for Signal (DOFS).**




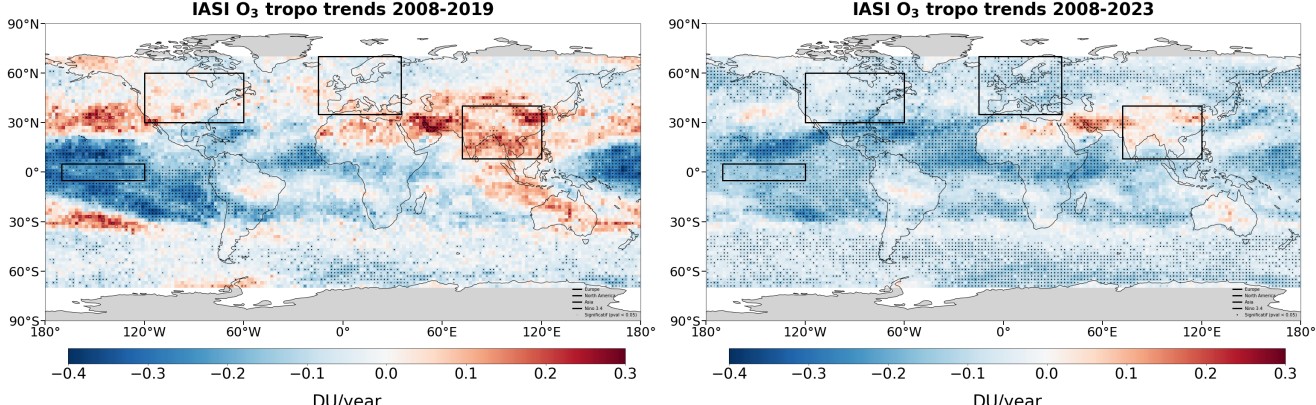

**Figure 12:** Spatial distribution of tropospheric ozone trends (in DU year$^{-1}$) derived from IASI-CDR before the COVID-19 pandemic (2008–2019; left panel) and including the pandemic (2008–2023; right panel). The tropospheric column is integrated from the surface to the thermal tropopause, as defined by the World Meteorological Organization (WMO, 1957). Dots indicate trends with 95% confidence. The black rectangles indicate the boundaries of the regions analyzed in Fig. 13.






**Figure 13: Monthly time series of tropospheric ozone anomalies (in DU) derived from IASI-CDR for different regions. The trend, uncertainty and p-value for the period before the COVID-19 pandemic (2008–2019) and including the pandemic (2008–2023) are indicated on each plot in green and red, respectively. Monthly time series of El Nino 3.4 index is also plotted in gray. The tropospheric column is integrated from the surface to the thermal tropopause, as defined by the World Meteorological Organization (WMO, 1957).**





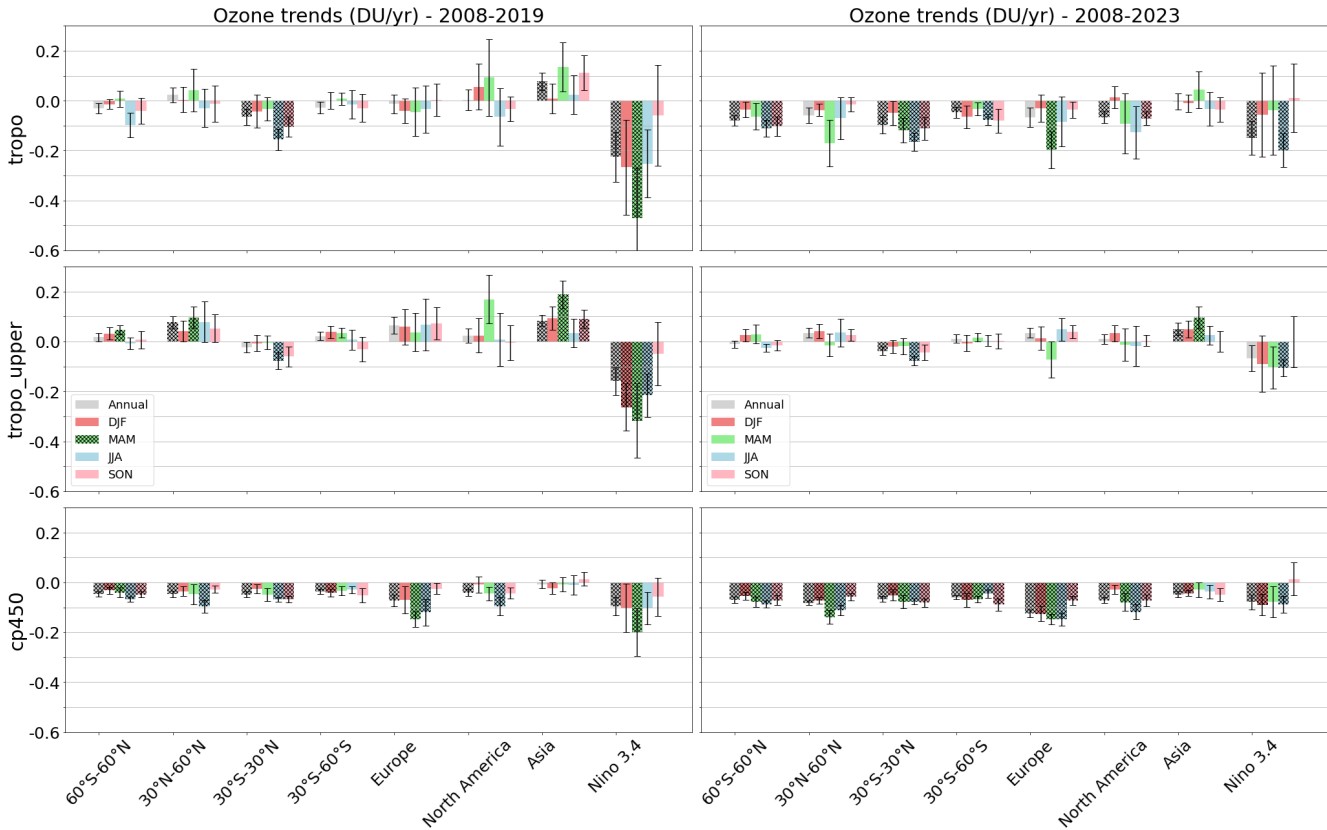


**Figure 14: Tropospheric ozone seasonal trends (in DU year[-1]) derived from IASI-CDR for different subcolumns (full, lower and upper tropospheric column) before the COVID-19 pandemic (2008–2019) and including the pandemic (2008–2023). The full, lower and upper tropospheric columns are defined as follows: surface to thermal tropopause, surface to 450 hPa, and 450 hPa to thermal tropopause, respectively. The thermal tropopause is estimated using the**

**World Meteorological Organization thermal definition (WMO, 1957). Hatchings indicate trends with 95% confidence.**






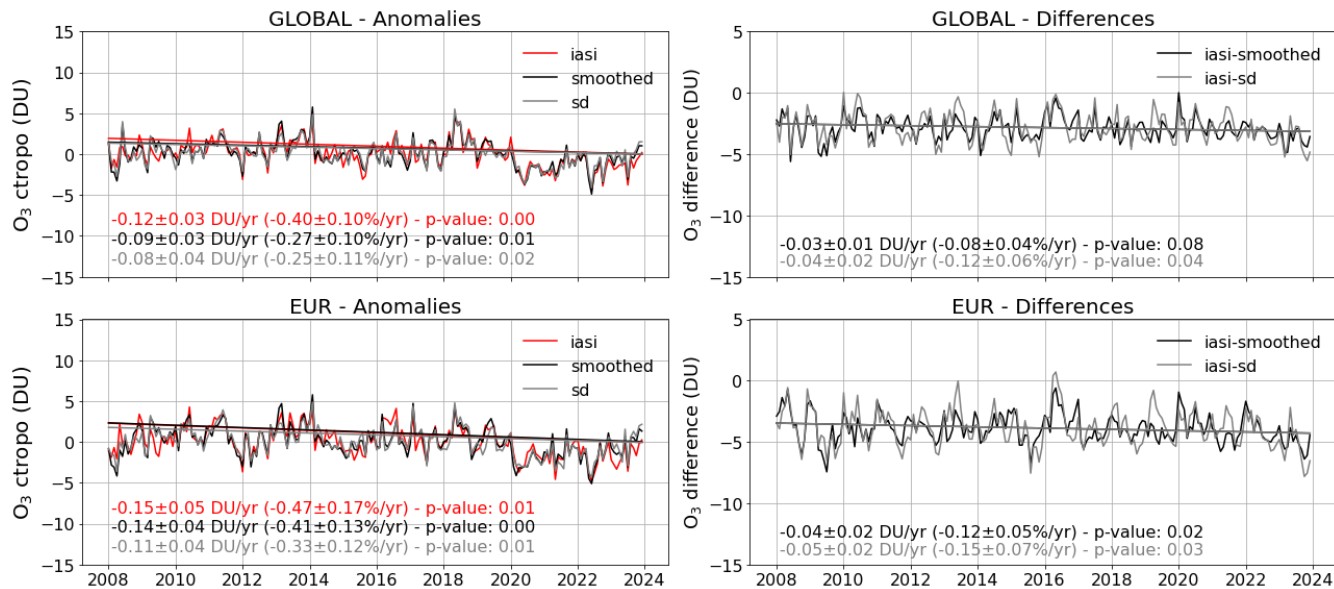

**Figure 15: Left: Monthly time series of tropospheric ozone anomalies in Dobson units (DU) derived from IASI CDR, smoothed and raw sonde data for the global scale and Europe. The tropospheric column is integrated from the surface to the thermal tropopause, as defined by the World Meteorological Organization (WMO, 1957). Right:**
**Corresponding monthly time series of tropospheric ozone difference between IASI and sonde data in DU. The black curve corresponds to the difference between IASI and smoothed sonde data and the gray curve corresponds to the difference between IASI and raw sonde data. The trend/drift, uncertainty and p-value for the period 2008–2023 are indicated on each plot.**







**Figure 16: Same as Fig. 15, but for individual sonde stations.**
