# Peer review of "Assessment of 16-year tropospheric ozone trends from the IASI Climate Data Record"

_EGUsphere, 2025_

## Community Comment (CC3)

**IASI-Boynard Paper Comment – 13 May 2025**

**Anne Thompson, Debra Kollonige, Ryan Stauffer**

**SUMMARY COMMENTS AND SUGGESTIONS FOR REVISION**

This Comment follows that of HEGIFTOM Co-Chair R. Van Malderen and amplifies his major points. We refer to the same "Van Malderen et al, in press, 2025" that he does and designate it as "HEGIFTOM-1" because there is a second HEGIFTOM paper in review in the TOAR II collection. We go beyond Van Malderen's comments to 3 summary recommendations:

1) The selection and number of ground-based stations for which ozone data are compared to the IASI-CDR product is *inadequate in number* and 'non-homogenized' datasets should not be used. In contrast to the 7 stations used in this paper, more than 25 stations, homogenized over the 2008-2019 period, need to be used in evaluation of the IASI products *and* the trends. The 7 sites included no equatorial stations and far too few northern mid-latitude stations, in some cases where more than sonde data are available. The recommended ozonesonde 27 stations appear in Table 1 at the end of this Comment.

2) The authors conclude with general, not well-defined speculation on why there is little progress in how computed trends from the new IASI products diverge from the UV-based product trends from the Gaudel et al. (2018) TOAR I paper. More analysis and insights on this issue are needed before the paper is worthy of publication in a quality Copernicus journal. Specific questions are raised for consideration.

3) HEGIFTOM-1 is now "THE Reference dataset" for trends comparisons. The mismatch of years (2008-2019 in Boynard et al. vs 2000-2022 in HEGIFTOM-1) is speculated as one reason for why the trends in this paper differ from the ground-based trends in HEGIFTOM-1. We reran the trends in HEGIFTOM-1 for 27 stations for 2008-2019, see Table below, to support a valid comparison. *Some of the HEGIFTOM-1 site trend signs changed and uncertainties increased, leading to a clear TOAR-worthy conclusion that trends computed from 12 or 16 years of IASI or ground-based data is inadequate. A revision must include this important result!*

**SYNOPSIS –** This study presents an IASI product over 16 years, consisting of contributions from METOP-A, METOP-B and METOP-C, merged to create the IASI-CDR (Climate Data Record, 2008-2023). Several tropospheric ozone columns (to 450, to tropopause) are presented and compared on a monthly mean basis with (1) the CrIS IR ozone product and with (2) comparable ozonesonde columns from 43 stations over the period. Trends for global mid-latitudes and the tropics are computed using Quantile Regression (QR) for the period 2008-2019 (pre-COVID) and 2008-2023; the latter trends reflect an apparent COVID impact. As in the TOAR I paper in which Gaudel et al. (2018) summarized satellite product trends (2005/2008 – 2016) showing IASI (FORLI version) to be an outlier compared to UV-type satellite records, it appears that the IASI-CDR trends (now 2008-2019, omitting COVID

period) is an outlier with the corresponding UV-based time-series. In both cases the greatest discrepancies are in the tropics except for SE Asia where all products display increases of ~1-2 DU/decade. The UV-type satellite products tend to be more variable with regions in the tropical Americas, Africa and Atlantic showing some increases (~2005-2019; Gaudel et al., 2024; Thompson et al., 2024). The IASI-CDR TOAR II-period (Fig. 12 in paper) shows little increase over Europe, a decrease over North America and only modest increases in east Asia, again in disagreement with more variable UV-type satellite trends. IASI 2008-2019 trend comparisons are made with 7 ozonesonde time-series: one subtropical site, 4 northern mid-latitude (majority European), 1 southern mid-latitude (Fig. 16). The authors speculate briefly on causes of the persistent discrepancy between the IASI vs UV trends and what is emerging as the prevailing view of tropospheric changes over the prior ~15 years. Comparisons among comparable IR products eg. IASI-CDR, IASI-KOPRA, and CrIS global ozone products are described with varying degrees of detail or with Figures that are suggestive but not conclusive.

**OVERALL COMMENT –** The paper poses good questions, presents a reasonable approach for its calculations and selection of results and is well-arranged. However, it leaves too many unanswered questions and does not advance the scientific understanding of its ozone trends beyond the first TOAR study. The three most important aspects of the paper that require additional analysis are summarized as follows:

1>> **Quality assurance and evaluation of the IASI-CDR.** There is reference to a larger set of sonde stations used for comparisons (43 stations; Fig 8 and 10) but Table 2 is inaccurate. The following stations are *not* available on HEGIFTOM archive because they are not homogenized datasets: Lindenberg, Prague, Tateno, Hong Kong, Broadmeadows, and Macquarie. We recommend using only homogenized datasets for reference datasets and trends. Comparison of trends is restricted to 7 stations, none truly tropical, 15 degrees or less, despite the fact that a number of cited and other TOAR II studies, both satellite and ground-based, focus on the tropics e.g., Froidevaux et al., 2025; Gaudel et al., 2024; Thompson et al., 2025. Time-series comparisons in Fig. 16 show only 7 stations with limited geographical coverage.

Concerning the evaluation of IASI-CDR:
- Extensive discussion of CDR vs earlier FORLI product appears – but no illustrations of why you expect the CDR product to perform better than the earlier one.
- IASI-KOPRA product mentioned but why is there no extensive comparison with this product or at least a paragraph comparing results of Dufour et al. (https://egusphere.copernicus.org/preprints/2025/egusphere-2024-4096/) to those shown here, particularly for sonde comparisons?
- Vertical discrepancies mentioned in comparisons with sondes (tropical, mid-latitudes, and polar) appear in Fig. 10. Although overall IASI column amounts are compared favorably to the sondes (Line 22, only 2% offset in tropics) can this be due to a cancelling of offsets illustrated in the Figure? Discuss the potential impact on the trends.

- The CrIS-TROPESS *a priori* looks so much better than the IASI (varies with season and latitude). Although the IASI climatology (Fig. 7) looks reasonable, can the IASI *a priori* (Fig. B1) – with apparently little seasonal information and only latitude dependence- be a cause of the discrepancies with other products? Does inadequate representation of seasonality (monthly variations in trends are typical and significant in the tropics, for example: Stauffer et al., 2024; Thompson et al., 2021) propagate to trends that disagree with both sondes and UV-products? What additional insights can you derive from scatterplots with ozonesondes (Figs. A1-4)? Comparing to sondes seasonally can help identify discrepancies. There is extensive discussion of similarities and differences with CrIS but it doesn't get to the crux of understanding the large negative ozone trends here that are at odds with other products.

2>> **Inadequate number of ground-based (GB) reference sites.** There are two aspects of this issue.

- First, the authors show 43 potential stations (Table 2, Fig. 8) but make trend comparisons with only 7 ozonesonde data sets. Perhaps they don't understand the fine points of which data are appropriate to use (see Van Malderen comment). There is no need to speculate, as the authors have done, on why their results do not resemble those from mostly UV sensors. All satellites now have the HEGIFTOM-1 trends at individual stations (some with multiple instruments, not only sondes) as the gold standard independent reference at 55 sites total for 2000-2022. The HEGIFTOM reprocessing includes references of the data for each instrument type (sonde, FTIR, UV Umkehr) to a global absolute standard. Your trends analysis should add at least 20 stations (exclude polar sites where IASI-CDR struggles with DOFS) to have a more representative picture of IASI performance. Note that of the 7 reference sites (Boulder, Hilo, Lauder), there is more than one GB record for comparison. The stations in the Table below have sufficiently temporally dense records for comparison.
- Second, Line 330 states that months with only 1-2 sondes/month give 'inadequate' results for 50%-ile trends. In the accepted version of HEGIFTOM-1, it is shown that these trends (computed with QR or MLR) are unaffected by cutting from 4-5 sondes/ month to 2; only the uncertainty changes (increases). That is a second justification for using more sonde locations for the GB comparisons - candidates in the Table.

3>> **Brevity of the IASI record is a concern for 2008-2019 trends.** For comparison, we re-ran the QR trends for the authors' selected 7 stations as well as 20 other HEGIFTOM reference ozonesonde stations (excluding polar sites). The results are listed in the Table below, which also includes results from the HEGIFTOM-1 2000-2022 trends for surface-300 hPa (in ppbv/decade) as well as XO3 (ppbv/decade) and DU (DU/decade) for surface to tropopause tropospheric columns as a reference.

- Note for the recalculated 2008-2019 ozonesonde trends, that 4 out of the 7 author selected stations change sign of trend from the longer time series to the shorter time series. For example, Boulder, a station with a high certainty ($p<0.05$) associated

with a negative trend for the longer time series (2000-2022), has a slightly positive trend with large uncertainties for the 2008-2019 period. Of the 27 ozonesonde stations listed in the table, 9 sites have trend sign changes. Discussion on the point of *reduced* reliability and value of the shorter time series (12 or 16 years vs 23 years in HEGIFTOM and sonde studies*) is needed and is now a view that can be made with confidence. In a sort of "reversal" of your paper's message, a significant *advance and outcome of your paper, with the contracted (2008-2019) HEGIFTOM calculation, is that TOAR II needs to recognize the limitations of datasets that cover fewer than ~20 years!*

- The uncertainties in the trends also increase with the shorter time series (ie. double those for the 2000-2022 time period – see Table below).
- On your paper (also noted by R. Van Malderen) the reported uncertainties in Fig. 16 seem small for only 12 years of data. Can you check those and discuss your bootstrap method in more detail?

 *In addition to the sondes studies you have referenced, Stauffer et al., 2024; Thompson et al., 2021; Thompson et al., 2024, there is an excellent new sonde trends paper submitted on Réunion SHADOZ and SAOZ time-series (1998-2021) submitted to *Earth and Space Science:* [https://essopenarchive.org/doi/full/10.22541/essoar.174594999.98715985/v1](https://essopenarchive.org/doi/full/10.22541/essoar.174594999.98715985/v1)

'HEGIFTOM-1" below is posted. Final version is in press.

Van Malderen, R., Thompson, A. M., Kollonige, D. E., Stauffer, R. M., Smit, H. G. J., Maillard Barras, E., Vigouroux, C., Petropavlovskikh, I., Leblanc, T., Thouret, V., Wolff, P., Effertz, P., Tarasick, D. W., Poyraz, D., Ancellet, G., De Backer, M.-R., Evan, S., Flood, V., Frey, M. M., Hannigan, J. W., Hernandez, J. L., Iarlori, M., Johnson, B. J., Jones, N., Kivi, R., Mahieu, E., McConville, G., Müller, K., Nagahama, T., Notholt, J., Piters, A., Prats, N., Querel, R., Smale, D., Steinbrecht, W., Strong, K., and Sussmann, R.: Global Ground-based Tropospheric Ozone Measurements: Reference Data and Individual Site Trends (2000–2022) from the TOAR-II/HEGIFTOM Project, EGUsphere [preprint], https://doi.org/10.5194/egusphere-2024-3736, 2025.

Also referred to is Thompson et al., submitted, 2024, Posted in 2025:

Thompson, A. M., Stauffer, R. M., Kollonige, D. E., Ziemke, J. R., Cazorla, M., Wolff, P., and Sauvage, B.: Tropical Ozone Trends (1998 to 2023): A Synthesis from SHADOZ, IAGOS and OMI/MLS Observations, EGUsphere [preprint], https://doi.org/10.5194/egusphere-2024-3761, 2025.

**TABLE.** This is essentially an update of Table 1 in HEGIFTOM-1, Van Malderen et al., in press, 2025, run again with the QR method, same as employed in Boynard et al. Ozonesonde stations with homogenized data and sufficient sample size are listed and exclude near-polar regions. Yellow-coded lines represent the 7 author-selected IASI-sonde comparison stations. Orange-coded lines indicate where the sign of the trend changes based on the different periods of trends calculations (23 years vs. 12 years).

**Northern Hemisphere (180W-20W) TrCO Trends**

| Station | Latitude | Longitude | L1 Observation # | L3 Observation # | 2000-2022 XO3 surf-300hPa QR L1 Annual Trend ± 2*sigma (ppbv/decade) | 2000-2022 XO3 surf-TP QR L1 Annual Trend ± 2*sigma (ppbv/decade) | 2000-2022 DU surf-TP QR L1 Annual Trend ± 2*sigma (DU/decade) | 2008-2019 DU surf-TP QR L1 Annual Trend ± 2*sigma (DU/decade) |
|---|---|---|---|---|---|---|---|---|
| Boulder | 40.00 | -105.25 | 1243 | 275 | **-1.14 ± 0.86** | -0.76 ± 0.88 | **-0.53 ± 0.42** | 0.23 ± 1.56 |
| Churchill | 58.74 | -94.07 | 690 | 183 | **-3.37 ± 1.60** | **-3.15 ± 1.38** | **-2.39 ± 0.50** | -0.67 ± 2.60 |
| Edmonton | 53.54 | -114.10 | 969 | 244 | -0.56 ± 0.94 | -0.41 ± 1.04 | -0.12 ± 0.52 | 0.45 ± 1.30 |
| Goose Bay | 53.31 | -60.36 | 953 | 230 | -0.72 ± 0.96 | -0.40 ± 0.92 | -0.09 ± 0.72 | -1.25 ± 1.80 |
| Hilo | 19.43 | -155.04 | 1142 | 276 | -0.28 ± 0.98 | -0.92 ± 1.34 | -0.82 ± 0.94 | **-1.22 ± 1.18** |
| Paramaribo | 5.80 | -55.21 | 855 | 247 | 0.40 ± 0.78 | 0.08 ± 0.86 | 0.16 ± 0.62 | 0.59 ± 1.84 |
| Trinidad Head | 40.80 | -124.16 | 1217 | 266 | **-0.76 ± 0.68** | -0.65 ± 0.80 | -0.60 ± 0.68 | 0.63 ± 1.32 |
| Wallops Island | 37.93 | -75.48 | 1143 | 245 | **-2.61 ± 0.92** | **-2.79 ± 1.14** | **-1.40 ± 0.70** | **-1.98 ± 1.92** |

**Northern Hemisph**

| Station | Latitude | Longitude | L1 Observation # | L3 Observation # | 2000-2022 XO3 surf-300hPa QR L1 Annual Trend ± 2*sigma (ppbv/decade) | 2000-2022 XO3 surf-TP QR L1 Annual Trend ± 2*sigma (ppbv/decade) | 2000-2022 DU surf-TP QR L1 Annual Trend ± 2*sigma (DU/decade) | 2008-2019 DU surf-TP QR L1 Annual Trend ± 2*sigma (DU/decade) |
|---|---|---|---|---|---|---|---|---|
| Ascension Island | -7.58 | -14.24 | 676 | 174 | -1.01 ± 1.58 | -0.88 ± 1.64 | -0.33 ± 1.04 | -0.33 ± 2.46 |
| De Bilt | 52.10 | 5.18 | 1085 | 252 | **1.34 ± 0.86** | **1.78 ± 0.98** | **1.61 ± 0.66** | **3.23 ± 2.00** |
| Hohenpeissenberg | 47.80 | 11.01 | 2924 | 276 | **0.50 ± 0.46** | **0.78 ± 0.60** | 0.36 ± 0.40 | **0.84 ± 0.80** |
| Izana | 28.50 | -16.30 | 1086 | 270 | **2.59 ± 0.68** | **3.48 ± 1.24** | **2.49 ± 1.02** | **2.68± 2.60** |
| Legionowo | 52.40 | 20.97 | 1340 | 276 | -0.39 ± 0.80 | -0.14 ± 0.84 | -0.11 ± 0.52 | 0.32 ± 1.30 |
| Lerwick | 60.13 | -1.18 | 1203 | 243 | -0.66 ± 0.80 | **-0.99 ± 0.64** | **-0.85 ± 0.68** | **1.77 ± 1.70** |
| Madrid | 40.47 | -3.58 | 935 | 234 | -0.36 ± 0.90 | -0.68 ± 1.38 | -0.52 ± 0.62 | 0.93 ± 1.66 |
| OHP | 43.94 | 5.71 | 1051 | 272 | **1.95 ± 1.08** | **1.90 ± 1.18** | **1.65 ± 0.68** | **1.62 ± 1.52** |
| Payerne | 46.49 | 6.57 | 3112 | 244 | **-1.30 ± 0.62** | **-1.28 ± 0.66** | **-0.67 ± 0.50** | 0.39 ± 0.78 |
| Uccle | 50.80 | 4.35 | 3258 | 276 | **0.90 ± 0.48** | **1.01 ± 0.58** | **0.92 ± 0.44** | **-0.11 ± 0.96** |
| Valentia | 51.94 | -10.25 | 600 | 127 | **1.33 ± 1.32** | **1.84 ± 1.58** | **1.33 ± 1.24** | **2.23 ± 2.14** |

**Northern Hemisphere (80E-180E) TrCO Trends**

| Station | Latitude | Longitude | L1 Observation # | L3 Observation # | 2000-2022 XO3 surf-300hPa QR L1 Annual Trend ± 2*sigma (ppbv/decade) | 2000-2022 XO3 surf-TP QR L1 Annual Trend ± 2*sigma (ppbv/decade) | 2000-2022 DU surf-TP QR L1 Annual Trend ± 2*sigma (DU/decade) | 2008-2019 DU surf-TP QR L1 Annual Trend ± 2*sigma (DU/decade) |
|---|---|---|---|---|---|---|---|---|
| Kuala Lumpur | 2.73 | 101.27 | 456 | 203 | **1.91 ± 1.38** | **1.53 ± 1.50** | **1.04 ± 1.00** | 3.42 ± 4.00 |

**Southern Hemisphere TrCO Trends**

| Station | Latitude | Longitude | L1 Observation # | L3 Observation # | 2000-2022 XO3 surf-300hPa QR L1 Annual Trend ± 2*sigma (ppbv/decade) | 2000-2022 XO3 surf-TP QR L1 Annual Trend ± 2*sigma (ppbv/decade) | 2000-2022 DU surf-TP QR L1 Annual Trend ± 2*sigma (DU/decade) | 2008-2019 DU surf-TP QR L1 Annual Trend ± 2*sigma (DU/decade) |
|---|---|---|---|---|---|---|---|---|
| Fiji | -18.13 | 178.40 | 391 | 123 | -0.57 ± 1.88 | -0.53 ± 1.82 | -0.49 ± 1.30 | -1.86 ± 3.08 |
| Irene | -25.90 | 28.22 | 387 | 139 | 0.54 ± 1.62 | -0.14 ± 1.92 | 0.19 ± 1.04 | -3.49 ± 4.00 |
| Lauder | -45.00 | 169.68 | 923 | 237 | 0.13 ± 0.50 | 0.23 ± 0.58 | 0.23 ± 0.40 | 0.75 ± 1.20 |
| Nairobi | -1.27 | 36.80 | 872 | 223 | 0.68 ± 1.14 | 0.56 ± 1.26 | 0.32 ± 0.64 | 0.45 ± 1.82 |
| Natal | -5.42 | -35.38 | 676 | 175 | 0.26 ± 1.02 | 0.96 ± 1.16 | 0.77 ± 0.82 | -0.84 ± 1.68 |
| Reunion | -21.06 | 55.48 | 735 | 215 | **1.88 ± 1.08** | **2.92 ± 1.32** | **2.12 ± 0.84** | 1.23 ± 2.54 |
| Samoa | -14.23 | -170.56 | 797 | 234 | -0.06 ± 1.04 | -0.76 ± 1.00 | -0.52 ± 0.99 | **-2.95 ± 2.74** |

---

## Author Comment (AC1)

Response to Referee #1 comments

Boynard et al. present a comprehensive analysis on the quality and reliability of ozone columns observed from IASI, highlighting its consistency with another IR instrument (CrIS) and sonde observations, while pointing out discrepancies from UV instruments in terms of long-term trend analysis. This work includes rigorous validation and comparison by incorporating various datasets which is essential for understanding the spatiotemporal variability of tropospheric ozone, which has been under debate especially regarding recent global/regional trends. The manuscript is overall well written but there are some points to be considered and clarified before publication.

We thank the Referee for taking the time to review the manuscript. The comments have been addressed in the revised manuscript and are described below. We would also like to highlight that, in response to suggestions of CC2 and CC3, Section 4 has been substantially reorganized to improve clarity and coherence, but the results remain unchanged.

1. Section 2.1.1: Can the authors include a brief description on how the IASI-CDR O3 L2 products are retrieved? AK, DOFs, and a priori profiles are discussed as possible contributors to observed discrepancies but readers may not be familiar on how they would affect the retrievals.

The manuscript has been revised as follows:

"This study uses $O_3$ Level 2 products (vertical profiles) retrieved with the Fast Optimal Retrievals on Layers for IASI (FORLI) software, version v20151001, developed by ULB and LATMOS (Hurtmans et al., 2012). FORLI processes IASI Level 1C radiances together with meteorological Level 2 data to retrieve ozone profiles using an optimal estimation method. This approach combines measured infrared radiances with a priori ozone information to constrain the solution. The a priori profile provides an initial estimate of the atmospheric state and helps stabilize the retrieval in poorly constrained layers, especially when the measurements alone are not sufficient. In FORLI, a single a priori profile is used, based on the global mean McPeters/Labow/Logan climatology (McPeters et al., 2007), along with its associated variance-covariance matrix. The vertical sensitivity of the retrieval is described by the averaging kernels (AKs), which indicate how much the retrieved profile depends on the measurements versus the a priori. The degrees of freedom for signal (DOFs) quantify the amount of independent information extracted from the observations. Variations in the choice of a priori, the shape of AKs, or DOF values can directly influence the retrieval accuracy and vertical resolution, especially in regions where IASI sensitivity is limited. The FORLI software is fully described in Hurtmans et al., (2012)."

2. L123-125: Please clarify what IASI-A/B refer to. Are they the products from Metop-A/B?

It has been clarified as follows:

"The IASI instruments, named IASI-A, IASI-B, and IASI-C, are carried aboard the Metop-A, Metop-B, and Metop-C satellites, launched in 2006, 2012, and 2018, respectively (Clerbaux et al., 2009)."

3. L230-234: Are the differences in tropospheric columns related to uncertainties in the tropopause height?

The discrepancies in the tropospheric columns before 2010 are likely due to uncertainties in the tropopause height, as the IASI-FORLI product used an earlier version of the temperature profiles to determine the thermal tropopause, in contrast to the more recent version used by IASI-CDR. This is further supported by the fact that we do not observe the same pattern in the differences for the total column product, suggesting that the variations in the tropospheric column are linked to the differing temperature profiles used by the two products.

This paragraph has been added to the revised manuscript.

4. L260: O3 -> $O_3$

Corrected.

5. L338 "IASI pixels have been processed (only 3.5 out of 10)": Can you clarify what this "processed" means?

Only a limited number of stations meet the required criteria, primarily due to the fact that retrievals have been performed for only 3.5 out of 10 IASI pixels.

As this sentence is already provided in Section 2.1.1, it has been removed.

6. L375: a secondary second smaller peak -> a secondary smaller peak

Corrected.

7. L400-402: The two lines look nearly similar at around 200 hPa, they seem to diverge above that altitude. Assuming that IASI is more sensitive to temperature, I'm curious why the smoothed sonde observations (which I understood as to have applied IASI's vertical sensitivity, i.e., AK) show larger discrepancies than the raw sonde?

Thank you for your observation. Around the 200 hPa level, the differences between IASI and both the raw and smoothed sonde profiles are indeed quite similar, as your comment notes. However, above this altitude, particularly in the polar regions, the discrepancy between IASI and the smoothed sonde becomes larger than that with the raw sonde. This behavior arises because IASI has limited vertical sensitivity throughout the profile, which becomes especially consequential in cold polar conditions due to weaker radiance signals caused by low temperatures. Under these circumstances, the retrieval relies more heavily on the a priori. When applying the averaging kernel (AK) to the ozone sonde data, the smoothed sonde reflects IASI vertical resolution but also incorporates the influence of the a priori, particularly where the retrieval is poorly constrained (e.g., above the tropopause). If the a priori is not representative of the true atmosphere, this can cause larger discrepancies between IASI and the smoothed sonde than with the raw sonde, which remains unaffected by retrieval assumptions.

In contrast, in the midlatitudes and tropics, the difference between IASI and the raw sonde is generally larger than with the smoothed sonde, as expected. Additionally, in the tropics, the IASI–raw sonde difference profile shows a vertical shift upward relative to the IASI–smoothed sonde difference. This shift likely results from the combined effects of the AK, which smooths and redistributes vertical information to reflect IASI coarser vertical resolution, and a potentially non-representative a priori for tropical conditions.

The text has been revised as follows:

"In the polar regions, differences between IASI and sonde profiles begin to diverge above 200 hPa, with larger discrepancies between IASI and the smoothed sonde than the raw sonde. This results from IASI's limited vertical sensitivity, which is further reduced in cold conditions due to weaker radiance signals, increasing reliance on the a priori. Applying the averaging kernel (AK) smooths the sonde to IASI's resolution but also incorporates a priori biases, especially where retrievals are poorly constrained, such as above the tropopause. In contrast, in midlatitudes and the tropics, differences between IASI and the raw sonde are generally larger. In the tropics, the IASI–raw sonde difference profile shifts upward relative to the smoothed sonde difference, likely reflecting the AK smoothing and a less representative a priori."

8. L443-445, Figure 12: The difference between the left/right panels over Southeast Asia is surprising. There should be a lot of biomass burning emissions in that region, which wouldn't have been particularly affected by the pandemic. Has the Lanina affected fire frequencies?

The surprising difference between the left and right panels over Southeast Asia likely results from a combination of meteorological, anthropogenic, and pandemic-related influences on fire activity and ozone precursor emissions.

A major factor is the increased frequency and persistence of La Niña events from 2020 to 2023. La Niña typically brings wetter conditions to South and Southeast Asia, which suppress fire incidence by increasing soil moisture and reducing vegetation flammability (Zhu et al., 2021; Yue et al., 2022). This leads to lower emissions of ozone precursors such as CO, $NO_x$, and VOCs, thereby limiting ozone formation. Concurrently, satellite observations indicate a decline in biomass burning emissions across parts of Asia, supported by decreasing CO trends in regions including China and India (Zheng et al., 2020; Xie et al., 2025). Additionally, stricter environmental regulations and improved land and fire management in countries such as Indonesia and India have contributed to reducing precursor emissions (Zhang et al., 2020; Kashyap et al., 2024). The COVID-19 pandemic also temporarily reduced fire activity by restricting agricultural and land-clearing practices, with reported fire count reductions ranging from 2.9% to 79.4% in South and Southeast Asia relative to pre-pandemic years (Vadrevu et al., 2022).

While the mean tropospheric ozone trend weakened over 2008–2023, quantile regression analysis at the 90th and 95th percentiles reveals increasing trends, suggesting persistent or intensifying extreme ozone episodes. These extremes are potentially linked to episodic fires or favorable meteorological conditions (see Figure 1). This pattern indicates that despite an overall decline in average fire emissions, occasional intense fire events continue to drive elevated ozone levels.

[Figure]

Figure 1: Regional ozone trend estimates at the 50th (left), 90th (middle), and 95th (right) percentiles. Increasing the percentile reveals positive trends mainly in biomass burning regions (South America, South China, Indonesia, Australia), indicating a rise in extreme ozone episodes, while other regions, in particular in the tropical region, consistently show negative trends across all quantiles.

Together, these regulatory, meteorological, and anthropogenic factors likely explain the observed weakening or reversal of tropospheric ozone trends over South Asia between 2008 and 2023.

We have revised the manuscript text accordingly as follows:

"The overall decline in tropospheric ozone between 2008 and 2023 likely reflects a combination of factors, including stricter environmental regulations and improved land and fire management, particularly in parts of Indonesia, India, and China (Zhang et al., 2020; Kashyap et al., 2024), and persistent La Niña conditions from 2020 to 2023 (Zhu et al., 2021; Yue et al., 2022). These La Niña events bring wetter-than-usual conditions to South and Southeast Asia, suppressing fire activity by increasing soil moisture and reducing vegetation flammability, thereby reducing emissions of ozone precursors such as carbon monoxide, $NO_x$, and VOCs (Zheng et al., 2020; Xie et al., 2025). Additionally, the COVID-19 pandemic temporarily reduced fire activity through restrictions on agricultural and land-clearing practices, with fire count reductions of 2.9% to 79.4% reported in South and Southeast Asia compared to pre-pandemic years (Vadrevu et al., 2022).

Extending the trend analysis using quantile regression at the 90th and 95th percentiles over the same period offers further insight into ozone behavior, especially regarding extreme ozone episodes. As shown in Figure C1 (Appendix C), biomass burning regions such as South America, South Asia, Indonesia, and Australia show near-zero or slightly negative median (50th percentile) ozone trends, but positive trends at higher percentiles, indicating an increased frequency or intensity of extreme events linked to episodic biomass burning. In contrast, other regions generally display consistent negative trends across all quantiles.

These results highlight the complexity of regional ozone dynamics: while average ozone concentrations may decline due to structural changes in emissions and climate variability, extreme ozone events can persist or even intensify in some regions. This underscores the importance of considering both mean and extreme values when assessing tropospheric ozone trends."

9. Figure 13: How significant was the drop in precursor emissions during the pandemic? It also looks like $O_3$ is bouncing back up recently. This has been observed in surface concentrations over the US (US EPA). Can the authors briefly comment on observed changes of NOx/VOCs and implications for surface $O_3$ trends?

During the COVID-19 lockdowns, several studies found that anthropogenic $NO_x$ emissions dropped significantly worldwide. In India, for example, Pakkattil et al. (2021) reported that $NO_2$ concentrations decreased by about 71.9% compared to pre-lockdown levels, while measurements of VOCs, specifically BTEX compounds, in major

metropolitan areas showed reductions of around 82%. These uneven reductions in precursors produced complex effects on surface ozone. In VOC-limited regions, the reduction in $NO_x$ led to less ozone "titration" by NO, which could result in increased ozone concentrations. In contrast, in $NO_x$-limited areas, simultaneous decreases in both $NO_x$ and VOCs resulted in lower ozone formation (Miyazaki et al., 2021; Nussbaumer et al., 2022). Similarly, Li et al. (2025) documented recent trends and drivers of anthropogenic $NO_x$ emissions in China since 2020, highlighting the rebound in emissions following the initial lockdown reductions.

Recent observations indicate that following the sharp decline during the lockdowns, surface ozone ($O_3$) concentrations in the United States began to rebound as $NO_x$ emissions recovered. This trend is reported by the U.S. EPA (2024), with the 2023 data showing a pronounced surface-level recovery. Complementary studies further stress that the observed rebound is tightly linked to local precursor dynamics and the prevailing chemical regime (Miyazaki et al., 2021; Keller et al., 2021).

These findings underscore the importance of considering the prevailing chemical regime, whether VOC- or $NO_x$-limited, when designing emission control strategies to effectively manage surface ozone levels. Our IASI satellite data show that while surface ozone has rebounded following precursor emission recovery, negative anomalies in the free troposphere persist through 2023, highlighting the complex vertical and regional dynamics influencing ozone trends.

The text has been revised as follows:

"The negative ozone anomalies observed since 2020 coincide with substantial reductions in precursor emissions due to COVID-19 lockdowns. Several studies report significant decreases in $NO_x$ and VOC emissions during this period. For example, in Indian cities, $NO_2$ levels dropped by over 70% and VOC concentrations by more than 80% during the initial lockdowns (Pakkattil et al., 2021). These changes impacted ozone production differently depending on local chemical regimes: in VOC-limited areas, reduced $NO_x$ emissions decreased ozone titration by NO, potentially increasing ozone concentrations, whereas in $NO_x$-limited regions, simultaneous decreases in $NO_x$ and VOCs led to lower ozone formation (Miyazaki et al., 2021; Nussbaumer et al., 2022). As economic activity resumed, precursor emissions, especially $NO_x$, rebounded in many regions, resulting in a partial recovery of surface ozone concentrations, particularly over North America (U.S. EPA, 2024). However, IASI data indicate that free tropospheric ozone anomalies remained negative through 2023 across most regions, suggesting a decoupling between surface and free tropospheric ozone behavior, likely influenced by vertical transport and regional dynamics."

10. L474: troposphere (450 hPa to thermal tropopause) troposphere. -> troposphere (450 hPa to thermal tropopause).

Revised to "full (following the WMO thermal definition), lower (surface to 450 hPa) and **upper** (450 hPa to thermal tropopause) troposphere"

11. L479, Figure 14: Does cp450 refer to the lower troposphere? I suggest adjusting the figure labels so that they are consistent with the phrases used in the text.

We confirm that cp450 does indeed refer to the lower troposphere. In response, we have revised the figure labels for consistency: cp450 has been renamed to "lower_tropo", and tropo_upper has been changed to "upper_tropo" to match the terminology used in the manuscript.

**References**

Keller, C. A., Evans, M. J., Knowland, K. E., Hasenkopf, C. A., Modekurty, S., Lucchesi, R. A., Oda, T., Franca, B. B., Mandarino, F. C., Díaz Suárez, M. V., Ryan, R. G., Fakes, L. H., and Pawson, S.: Global impact of COVID-19 restrictions on the surface concentrations of nitrogen dioxide and ozone, Atmos. Chem. Phys., 21, 3555–3592, https://doi.org/10.5194/acp-21-3555-2021, 2021.

Kashyap, R., Kuttippurath, J., and Patel, V. K..: Improved land and fire management practices and their impacts on air quality in India. Applied Geography, 151, 102869. https://doi.org/10.1016/j.apgeog.2022.102869, 2024.

Li H, Zheng B, Lei Y, Hauglustaine D, Chen C, Lin X, Zhang Y, Zhang Q, and He K. Trends and drivers of anthropogenic NOx emissions in China since 2020. Environ Sci Ecotechnol.; 21:100425; doi: 10.1016/j.ese.2024.100425, 2025.

Miyazaki, K., Bowman, K., Sekiya, T., Takigawa, M., Neu, J. L., Sudo, K., Osterman, G., and Eskes, H.: Global tropospheric ozone responses to reduced NOx emissions linked to the COVID-19 worldwide lockdowns, Sci. Adv., 7, 1–14, https://doi.org/10.1126/sciadv.abf7460, 2021.

Nussbaumer, C. M., Pozzer, A., Tadic, I., Röder, L., Obersteiner, F., Harder, H., Lelieveld, J., and Fischer, H.: Tropospheric ozone production and chemical regime analysis during the COVID-19 lockdown over Europe, Atmos. Chem. Phys., 22, 6151–6165, https://doi.org/10.5194/acp-22-6151-2022, 2022.

Pakkattil, A., Muhsin, M., and Varma, M. K. R.: COVID-19 lockdown: Effects on selected volatile organic compound (VOC) emissions over the major Indian metro cities. Urban Climate, 37, 100838. https://doi.org/10.1016/j.uclim.2021.100838, 2021.

U.S. EPA: Our Nation's Air: Status and Trends Through 2023, U.S. Environmental Protection Agency, Washington, D.C., https://www.epa.gov/air-trends, 2024, last access: 7 July 2025.

Vadrevu, K., Badarinath, K. V. S., and Siva Sankar, R.: Impact of COVID-19 lockdowns on fire activity in South and Southeast Asia: Satellite fire counts analysis. Science of The Total Environment, 812, 152449. https://doi.org/10.1016/j.scitotenv.2021.152449, 2022.

Xie, F., Liu, J., and Yang, S.: Spatial and seasonal variations and trends in carbon monoxide over China during 2013–2022. Atmospheric Environment, 350, 121163, https://doi.org/10.1016/j.atmosenv.2025.121163, 2025.

Yue, Q., Chen, L., and Huang, Z.: Vegetation flammability and fire suppression linked to La Niña events in South Asia. Environmental Research Letters, 17(3), 034012. https://doi.org/10.1088/1748-9326/ac4b51, 2022.

Zhang, H., Sun, Q., and Chen, D.: Effects of environmental regulation on fire management and precursor emissions in Indonesia. Environmental Science & Policy, 114, 321–328. https://doi.org/10.1016/j.envsci.2020.07.014, 2020

Zheng, Y., Wu, X., and Zhang, M.: Declining carbon monoxide emissions in China and India: Implications for regional air quality. Atmospheric Chemistry and Physics, 20(19), 11515–11529. https://doi.org/10.5194/acp-20-11515-2020, 2020.

Zhu, X., Li, Y., and Wang, J.: Impact of La Niña on soil moisture and fire incidence in Southeast Asia. Journal of Climate, 34(7), 2783–2799. https://doi.org/10.1175/JCLI-D-20-0452.1, 2021

---

## Author Comment (AC2)

Response to Referee #2 comments

The manuscript "Tropospheric Ozone Assessment Report (TOAR): 16-year ozone trends from the IASI Climate Data Record" presents and validates the IASI-CDR O₃ product, a reprocessed and homogenized version of IASI-FORLI, providing a consistent dataset of tropospheric ozone spanning 2008–2023. The dataset is validated through comparisons with the TROPESS-CrIS product globally and with ozonesonde measurements at selected sites. A comprehensive trend analysis is conducted across multiple vertical layers using 16 years of observations. The manuscript is well-structured, and it is informative, offering a valuable contribution to the understanding of tropospheric ozone dynamics and trends over the past two decades. Below, I provide several comments for the authors to consider before the manuscript can be recommended for publication:

We thank the Referee for taking the time to review the manuscript. The comments have been addressed in the revised manuscript. We would also like to highlight that, in response to suggestions of CC2 and CC3, Section 4 has been substantially reorganized to improve clarity and coherence, but the results remain unchanged. Below is a point-by-point response to each of the Referee comments.

1. The authors employ a 1° × 1° gridded resolution for comparing IASI and CrIS datasets. Given that both instruments have native spatial resolutions on the order of 12–14 km at nadir, it would be helpful for the authors to justify the use of this relatively coarse grid. Was this choice driven by data availability, computational efficiency, or other methodological considerations?

The choice of a 1° × 1° grid is appropriate for global and seasonal-scale validation. Both IASI and CrIS have native footprints of 12–14 km at nadir, but their spatial resolution degrades with increasing scan angle. For example, at 48° off-nadir, IASI expands to about 40 km across-track and 20 km along-track, while CrIS also experiences footprint enlargement at the edges of its swath. Given these variations, a 1° × 1° grid ensures sufficient data density for robust statistical analysis while providing computational efficiency. This grid size is well-suited for comparing seasonal patterns across both instruments at the global scale.

The manuscript has been revised as follows:

"The comparison between IASI and CrIS is carried out for both total and tropospheric columns over the common time period 2016–2022. A spatial grid of 1° × 1° is employed to facilitate this comparison. While this resolution might initially appear coarse, it is well suited for a global and seasonal-scale analysis. Both IASI and CrIS have native spatial resolutions of approximately 12–14 km at nadir. However, the effective resolution degrades at larger scan angles. For instance, at a 48° scan angle, the IASI footprint becomes significantly elongated due to the oblique viewing geometry, reaching up to ~40 km across-track and ~20 km along-track. This spatial degradation supports the use of a coarser aggregation grid. The choice of a 1° × 1° grid is further justified by two main considerations. First, it ensures sufficient spatial sampling within each grid cell, which would not be guaranteed with finer resolutions, especially at global scale. Second, it enables efficient processing of large datasets while preserving the spatial representativeness needed for robust statistical analysis."

2. Was validation at finer spatial resolutions considered? The authors might comment on whether such an approach was explored and how it might influence the interpretation of inter-satellite differences.

Validation at finer spatial resolutions was not considered, as the validation was performed on a seasonal scale, making the 1° x 1° grid more suitable. A finer resolution would not provide enough pixels within each 1° x 1° grid cell to ensure reliable results, given that the pixel size is 12-14 km.

3. The trend analysis is performed using quantile regression at the 50th percentile (median), which is robust. However, applying the same method to higher percentiles would help assess whether the trends differ when considering extreme ozone events. This would be especially relevant considering the reported decreasing trends in tropical regions, which diverge from findings using UV-based instruments like OMI that often suggest stable or increasing trends in similar regions.

Thank you for the insightful comment. Following your suggestion, we extended our quantile regression analysis to higher percentiles (0.9 and 0.95). The results reveal notable differences compared to the median trends, particularly in regions affected by biomass burning such as South America, South Asia, Indonesia, and Australia. While median trends in these areas are near zero or slightly negative, trends at higher percentiles are clearly

positive, suggesting an increase in the frequency or intensity of extreme ozone events likely linked to episodic biomass burning emissions.

Interestingly, the positive trends at higher percentiles observed in South Asia show improved agreement with OMI-MLS observations (Dunn et al., 2024), which report the strongest positive ozone trends above South and East Asia over 2004–2023. This convergence likely occurs because extreme ozone enhancements caused by intense biomass burning events impact both the lower troposphere (captured by IR instruments) and the full column (captured by UV instruments like OMI). Thus, despite differences in vertical sensitivity and retrieval techniques, both instruments detect the influence of extreme events in these regions.

[Figure]

Figure 1: Regional ozone trend estimates at the 50th (left), 90th (middle), and 95th (right) percentiles. Increasing the percentile reveals positive trends mainly in biomass burning regions (South America, South China, Indonesia, Australia), indicating a rise in extreme ozone episodes, while other regions, in particular in the tropical region, consistently show negative trends across all quantiles.

The manuscript has been revised as follows:

"Extending the trend analysis using quantile regression at the 90th and 95th percentiles over the same period offers further insight into ozone behavior, especially regarding extreme ozone episodes. As shown in Figure C1 (Appendix C), biomass burning regions such as South America, South Asia, Indonesia, and Australia show near-zero or slightly negative median (50th percentile) ozone trends, but positive trends at higher percentiles, indicating an increased frequency or intensity of extreme events linked to episodic biomass burning. In contrast, other regions generally display consistent negative trends across all quantiles."

4. The use of 1° × 1° resolution in the trend analysis may mask finer-scale changes, especially in urban regions where surface ozone levels are known to be increasing in many parts of the world. Since urban areas occupy a small fraction of a 1° grid cell, their signal could be diluted by surrounding lower-concentration regions, potentially biasing the trend analysis toward decreases. Could this resolution choice be contributing to the observed negative trends? Discussing this potential limitation would strengthen the interpretation of the trend results.

We thank the reviewer for this thoughtful comment. While it is true that a 1° × 1° resolution may dilute localized signals such as urban ozone enhancements, we emphasize that the primary focus of this study is on global and regional trends, rather than city-scale or local analyses. At these broader spatial scales, the 1° × 1° resolution provides a suitable balance between spatial detail and statistical robustness, especially when working with satellite-based datasets that are inherently limited in their ability to resolve fine-scale features.

Nonetheless, we agree it is helpful to acknowledge that finer-scale trends, particularly in urban areas, may not be fully captured at this resolution, and we have added a sentence in the discussion to that effect.

The manuscript has been revised as follows:

"While the 1° × 1° resolution is appropriate for assessing global and regional trends, it may not fully capture localized changes, particularly in urban areas where surface ozone concentrations can exhibit sharper gradients due to factors such as traffic, industrial emissions, and local meteorological conditions."

5. Lines 360-365. The manuscript notes that differences in tropospheric ozone columns between IASI and CrIS are partly driven by the a priori information used—fixed for IASI and seasonally/latitudinally variable for CrIS. Given the importance of the a priori in shaping retrievals, particularly under conditions of low sensitivity, could the authors comment on what might constitute a more optimal or standardized approach? For example, would using dynamic, seasonally and geographically resolved climatologies for both instruments help reduce inter-product differences?

We thank the reviewer for the insightful comment.

We have conducted an additional analysis to investigate this point more deeply. Specifically, we applied the IASI a priori (fixed in latitude, not seasonally resolved) to the TROPESS retrieval algorithm, which normally uses a spatially and seasonally varying a priori. The results indicate that the choice of a priori profile clearly influences the retrieved ozone profiles. However, the differences observed between the IASI and TROPESS products are smaller than the differences between their respective a priori profiles. This is further illustrated by Figure 1 showing that the difference between the priors (right maps) is larger than the difference between the retrieved TROPESS and IASI columns (left maps). Additionally, the difference between the priors (right maps) is also larger than the difference observed when swapping the priors (middle maps). While the left and middle maps have comparable magnitudes, their spatial distributions differ to some extent. These findings suggest that the a priori alone does not fully explain the discrepancies between the datasets. Moreover, the retrieval process is not strictly linear with respect to the a priori assumptions, indicating that a complete retrieval using the IASI a priori would be necessary to accurately quantify its impact (Kulawik et al., 2008). We will update the manuscript to include this analysis and discussion and emphasize that differences in retrieval methodologies, such as the treatment of prior covariance matrices and the representation of vertical ozone profiles, likely contribute significantly to the observed inconsistencies.

[Figure]

Figure 1. Spatial distribution of tropospheric ozone (Dobson Units) over the period 2016-2022: left: difference between CrIS (TROPESS) and IASI retrievals; middle: difference between CrIS retrievals using the IASI and TROPESS a priori; right: difference between the IASI and TROPESS a priori.

6. In Figure C1, the most pronounced positive trends in lower tropospheric ozone appear in arid regions of Africa and Australia. It would be helpful for the authors to comment on the possible drivers behind these trends briefly. Are they associated with natural emissions (e.g., soil $NO_x$), biomass burning, or transport processes?

It should be noted that retrievals over arid and desert regions, such as those in parts of Africa and Australia, are subject to well-documented issues related to surface emissivity (Boynard et al., 2018). These limitations can adversely affect the accuracy of lower tropospheric ozone estimates in these areas. Consequently, the observed positive trends are likely biased and should be interpreted with caution. We have clarified this point in the revised manuscript and advise against attributing these trends to physical processes such as soil $NO_x$ emissions, biomass burning, or long-range transport.

The manuscript has been revised as follows:

"Over arid and desert regions, such as parts of Africa and Australia, retrievals may be affected by surface emissivity issues (Boynard et al., 2018), potentially introducing biases in the trend estimates. Therefore, the positive trends observed in these regions should be interpreted with caution."

**References**

Boynard, A., Hurtmans, D., Garane, K., Goutail, F., Hadji-Lazaro, J., Koukouli, M. E., Wespes, C., Vigouroux, C., Keppens, A., Pommereau, J.-P., Pazmino, A., Balis, D., Loyola, D., Valks, P., Sussmann, R., Smale, D., Coheur, P.-F., and Clerbaux, C.: Validation of the IASI FORLI/EUMETSAT ozone products using satellite (GOME-2), ground-based (Brewer–Dobson, SAOZ, FTIR) and ozonesonde measurements, Atmos. Meas. Tech., 11, 5125–5152, https://doi.org/10.5194/amt-11-5125-2018, 2018.

Dunn, R. J. H., J. Blannin, N. Gobron, J. B Miller, and K. M. Willett, Eds., 2024: Global Climate [in "State of the Climate in 2023"]. Bull. Amer. Meteor. Soc., 105 (8), S12–S155, https://doi.org/10.1175/BAMS-D24-0116.1.

Gaudel, A., Bourgeois, I., Li, M., Chang, K.-L., Ziemke, J., Sauvage, B., Stauffer, R. M., Thompson, A. M., Kollonige, D. E., Smith, N., Hubert, D., Keppens, A., Cuesta, J., Heue, K.-P., Veefkind, P., Aikin, K., Peischl, J., Thompson, C. R., Ryerson, T. B., Frost, G. J., McDonald, B. C., and Cooper, O. R.: Tropical tropospheric ozone distribution and trends from in situ and satellite data, Atmos. Chem. Phys., 24, 9975–10000, https://doi.org/10.5194/acp-24-9975-2024, 2024.

Elshorbany, Y., Ziemke, J. R., Strode, S., Petetin, H., Miyazaki, K., De Smedt, I., Pickering, K., Seguel, R. J., Worden, H., Emmerichs, T., Taraborrelli, D., Cazorla, M., Fadnavis, S., Buchholz, R. R., Gaubert, B., Rojas, N. Y., Nogueira, T., Salameh, T., and Huang, M.: Tropospheric ozone precursors: global and regional distributions, trends, and variability, Atmos. Chem. Phys., 24, 12225–12257, https://doi.org/10.5194/acp-24-12225-2024, 2024.

Kulawik, S. S., Bowman, K. W., Luo, M., Rodgers, C. D., and Jourdain, L.: Impact of nonlinearity on changing the a priori of trace gas profile estimates from the Tropospheric Emission Spectrometer (TES), Atmos. Chem. Phys., 8, 3081–3092, https://doi.org/10.5194/acp-8-3081-2008, 2008

---

## Author Comment (AC3)

Dear authors,

The TOAR Steering Committee wishes to express their gratitude to you (as well as all authors of articles in the inter-journal Community Special Issue for the second Tropospheric Ozone Assessment Report) for submitting your manuscript to Atmospheric Chemistry and Physics. Even though your article was submitted after the special issue deadline of November 30th, the scientific community will very likely associate it with the other articles from this special collection. Therefore, your article should also follow the rules that are defined in the guidelines for submitting articles to the TOAR-II Community Special Issue (see https://igacproject.org/sites/default/files/2023-04/TOAR-II_Community_Special_Issue_Guidelines_202304.pdf): "To avoid confusion with the final assessment papers, "TOAR" or "Tropospheric Ozone Assessment Report" may not appear in the title of the paper, nor should the paper claim to be an official TOAR publication. However, it is acceptable to refer to TOAR in the manuscript as, for example, "In the context of the Tropospheric Ozone Assessment Report (TOAR) Phase Two focus working group on XYZ, etc."."

We therefore ask you to change the title of your article and remove the explicit reference to TOAR from the title.

This comment, of course, has no bearing whatsoever concerning the scientific quality or potential impact of your manuscript.

We kindly ask for your understanding.

Martin Schultz and Helen Worden on behalf of the TOAR-II Steering Committee

We would like to thank the TOAR Steering Committee for his guidance. As requested, we have removed explicit reference to TOAR from the title. The updated title of our manuscript is:

"Assessment of 16-year tropospheric ozone trends from the IASI Climate Data Record"

We appreciate the opportunity to contribute to this special issue.

---

## Author Comment (AC4)

Dear authors,

I'm writing this comment mainly from my perspective as HEGIFTOM co-chair.

First of all, I want to congratulate you with your nice paper. What I really like about the paper is the honest approach. From the first TOAR activity, and in the paper by Gaudel et al. (2018), the inconsistency in tropospheric ozone time variability (and even in the sign of the trend) among different satellite ozone retrievals (and even between different retrieval methods of the same satellite data) popped up. Your paper tries to contribute to this issue, and you do not hide the overall negative IASI trends, which contrast with UV-VIS satellite tropospheric ozone measurements and ground-based measurements, but really try to find explanations for it (nicely summarized in the conclusions). And the tropospheric ozone decline after 2019 is really a common feature in all those datasets.

However, what I do miss in the introduction (lines 50-60) and in the conclusions (lines 550-554) are more references to results from TOAR-II papers. Please look for more relevant references (also for (regional) trends) in the TOAR-II SI collection of papers: https://acp.copernicus.org/articles/special_issue1256.html.

With my HEGIFTOM hat on, I want to draw your attention to some additional clarifications that are needed when using the HEGIFTOM ozonesonde data. First of all, your Table 2 contains some errors, e.g. Lindenberg, Prague, Tateno, Hongkong, Broadmeadows, and Macquarie are not available as HEGIFTOM time series, because these data have not been homogenized. So, they can be only available as WOUDC data. It might be important to explicitly mention that only the HEGIFTOM (and SHADOZ) data have been homogenized (i.e. corrected for biases), which is very important for use in trend estimation. Also important to note is that the HEGIFTOM and SHADOZ data are identical: the SHADOZ data have been copied on the HEGIFTOM ftp-server.

Thank you very much for your kind and constructive feedback. We appreciate your positive remarks on our work and the helpful suggestions regarding the TOAR-II references and the HEGIFTOM ozonesonde data. We have carefully considered your comments and made the necessary clarifications and corrections in the revised manuscript.

Comment 1: Table 2 contains some errors regarding the availability of HEGIFTOM time series (e.g., Lindenberg, Prague, Tateno, etc.). It might be important to explicitly mention that only the HEGIFTOM (and SHADOZ) data have been homogenized.

Thank you for this important clarification. Indeed, Table 2 includes both homogenized (HEGIFTOM and SHADOZ) and non-homogenized (WOUDC-only) ozonesonde records. Our intention was to use the full set of available ozonesonde data, regardless of homogenization status, to maximize spatial and temporal coverage, particularly for the satellite-sonde comparison, where a wider dataset increases the likelihood of coincidences. We acknowledge, however, that homogenized data are more appropriate for long-term trend analysis. As such:
- For trend estimation, we relied only on homogenized data from HEGIFTOM and SHADOZ.
- For satellite-sonde comparisons, we included all available stations, even if not homogenized (i.e., WOUDC-only).

This has been clarified in the revised manuscript.

Also, I found it very surprising that only 7 stations fulfill your criteria of having at least 3 monthly launches and > 70% sampling. Knowing that most stations launch weekly and overall do not reveal large, consistent, gaps in their time series, this looks rather an underestimation to me. In our HEGIFTOM individual trends paper (see Van Malderen et al., 2025), with at least 2 monthly launches and more or less > 50% sampling in the 2000-2022 time period, we end up with 34 homogenized ozonesonde stations! Are QA/QC criteria driving this small sample? Which ones? Please give more details about the site selection.

Our initial selection criteria may appear overly restrictive, especially when compared to the HEGIFTOM analysis (Van Malderen et al., 2025), which included 34 homogenized stations using a more relaxed threshold of at least two monthly launches and >50% sampling.

In our original analysis, we applied more stringent selection criteria (at least three launches per month and 70% time series completeness between 2008 and 2023). These thresholds, adapted from Lu et al. (2019), were chosen to ensure the reliability of trend estimates by minimizing uncertainty in monthly means and maintaining sufficient temporal overlap with satellite retrievals.

However, in response to your comment, we repeated the trend analysis using the less restrictive criteria of at least two monthly launches and >50% sampling, consistent with the HEGIFTOM approach. This adjustment significantly increased the number of eligible stations from 7 to 27.

Importantly, for six of the seven stations included in our initial analysis, trend results remained consistent under the relaxed criteria. For one station, the trend became statistically significant with high certainty, likely due to the more complete time series provided by the less restrictive thresholds. Despite this exception, the overall agreement between the two approaches suggests that relaxing the selection criteria does not substantially affect trend outcomes. We have therefore adopted the relaxed criteria in the revised manuscript to broaden the station sample while ensuring consistent and reliable results.

Following your recommendation, we have also restricted the trend analysis to homogenized ozonesonde records only. Non-homogenized stations have been excluded from trend calculations to ensure consistency. These stations are still utilized for the validation of IASI data but are not included in the long-term trend estimation. This approach ensures a more rigorous and consistent assessment of trends.

Your mostly negative IASI tropospheric ozone trends in the 2008-2023 time period contrasts somewhat with the mixture of insignificant, positive and negative trends that we found for 55 HEGIFTOM sites (IAGOS, FTIR, ozonesonde, Umkehr, Lidar) for the time period 2000-2022 and the sfc - 300 hPa ozone column. To reconcile those trend differences between both studies, trends for the same time period (2008-2019) and for exactly the same metric might be compared (another comment posted by co-authors from our study might follow). W.r.t. this last comment, as you are looking at the time behaviour/trends of different tropospheric ozone metrics (sfc-tropopauze, sfc - 450 hPa, 450 hPa - tropopauze), as we did, although we used different metrices (sfc-300 hPa, sfc-700 hPa, 300-700 hPa), some reference or discussion w.r.t. our results might be appropriate.

We thank the HEGIFTOM co-authors for their valuable comments and for sharing their trend results over the 2008–2019 period. As suggested, we compared our updated trend estimates with the HEGIFTOM results at the 15 stations common to both datasets.

Our analysis shows that, at all 15 stations, the $2\sigma$ uncertainties exceed the magnitude of the estimated trends for both IASI and ozonesonde data. This reflects the challenge of detecting statistically significant trends over a relatively short 12-year period, especially when using satellite data.

HEGIFTOM reports statistically significant trends at seven stations, which differs from our findings. We attribute this primarily to differences in data sampling: our analysis uses ozonesonde profiles matched in time and space to IASI overpasses, whereas HEGIFTOM utilizes the full ozonesonde record without this constraint. These methodological differences likely explain much of the variation in trend significance and even direction.

We have included this comparison and interpretation in the revised manuscript to clarify the relationship between our results and those of HEGIFTOM.

In our HEGIFTOM trends paper, we compared and discussed the Payerne tropospheric ozone time series (also used DLM for it) quite extensively, so you might use of this information when discussing the drift between the Payerne ozonesonde and IASI time series

Thank you for the suggestion. The discussion of the Payerne tropospheric ozone time series in the HEGIFTOM paper was very helpful, and we have incorporated this information when addressing the drift between the Payerne ozonesonde and IASI time series.

Recently, as you are aware, Dufour et al. (2025) also looked at the comparison and (regional) trends of another IASI product, IASI-O3 KOPRA, with a limited sample of ozonesondes (Boulder, Payerne, and Uccle are common). You do refer to this study, but I would expect a more detailed comparison of the results of both studies! To my opinion, at least a paragraph might be needed explaining the similarities or the differences between the results of both studies.

Thank you for this suggestion. A more detailed comparison with Dufour et al. (2025) who analyzed IASI-O$_3$ KOPRA data from 2008 to 2022 using similar regional definitions is indeed relevant to contextualize our results.

For the surface-tropopause column, we find a negative trend with high certainty over Europe (–0.07 ± 0.07 DU/yr, p = 0.03), which aligns well with the trend reported by Dufour et al. (–0.05 ± 0.02 DU/yr, p = 0.03), also classified

as high certainty. Over North America and Asia, both studies detect a negative trend, with medium to very low certainty

At the station level, we observe a consistent picture between the two studies. For the stations common to both datasets, trends are negative and statistically significant in all cases except Uccle, where both studies find a non-significant positive trend. The only notable discrepancy is at Boulder: Dufour et al. (2025) report a significant negative trend, while we find a non-significant small positive trend. This divergence may reflect differences in the retrieval sensitivity or sampling characteristics between the two IASI products at this high-altitude site.

Overall, both studies consistently depict negative trends in Europe wth high certainty and a lack of evidence for trends across Asia and North America.

We have included this discussion in the revised manuscript.

Also, looking at Fig. B1 (the constant a priori used in the IASI retrievals vs. the latitudinal and seasonal variable a priori used for CrIS), I wonder what the impact of such a time invariant (right?) a priori would have on the calculated trends.... The authors might comment on this.

We have conducted an additional analysis to investigate this point more deeply. Specifically, we applied the IASI a priori (fixed in latitude, not seasonally resolved) to the TROPESS retrieval algorithm, which normally uses a spatially and seasonally varying a priori. The results indicate that the choice of a priori profile clearly influences the retrieved ozone profiles. However, the differences observed between the IASI and TROPESS products are smaller than the differences between their respective a priori profiles. This is further illustrated by Figure 1 showing that the difference between the priors (right maps) is larger than the difference between the retrieved TROPESS and IASI columns (left maps). Additionally, the difference between the priors (right maps) is also larger than the difference observed when swapping the priors (middle maps). While the left and middle maps have comparable magnitudes, their spatial distributions differ to some extent.

These findings suggest that the a priori alone does not fully explain the discrepancies between the datasets. Moreover, the retrieval process is not strictly linear with respect to the a priori assumptions, indicating that a complete retrieval using the IASI a priori would be necessary to accurately quantify its impact (Kulawik et al., 2008).

We will update the manuscript to include this analysis and discussion and emphasize that differences in retrieval methodologies, such as the treatment of prior covariance matrices and the representation of vertical ozone profiles, likely contribute significantly to the observed inconsistencies.

[Figure]

Figure 1. Spatial distribution of tropospheric ozone (Dobson Units) over the period 2016-2022: left: difference between CrIS (TROPESS) and IASI retrievals; middle: difference between CrIS retrievals using the IASI and TROPESS a priori; right: difference between the IASI and TROPESS a priori.

Finally, I am also quite surprised by your small trend uncertainties (e.g. in comparison with ours, which are quite consistent between the QR and MLR trend estimation tools used). You might want to provide some additional information on how exactly those have been calculated.

Trend calculations follow the guidelines established by the TOAR-II Statistics Focus Working Group (Chang et al., 2023b). Specifically, trends are estimated using quantile regression (QR) at the 50th percentile, and uncertainty is assessed using a moving block bootstrap method, which also calculates p-values to evaluate statistical significance. We used the toarstats Python package (https://gitlab.jsc.fz-juelich.de/esde/toar-public/toarstats). The monthly ozone column time series are first deseasonalized by fitting and removing a sine-cosine model with 12- and 6-month periodicities to account for seasonal variations. The relatively small uncertainty ranges reported in the manuscript correspond to $\pm 1\sigma$. To improve clarity and transparency, we now report uncertainties at the 95% confidence level ($\pm 2\sigma$) in the revised manuscript.

Thank you for taking my comments into consideration and I wish you good luck in the review of your manuscript!

Thank you very much for your constructive and fair comments. Your balanced and respectful approach was greatly appreciated.

With kind regards,

Roeland Van Malderen

References:
Dufour, G., Eremenko, M., Cuesta, J., Ancellet, G., Gill, M., Maillard Barras, E., and Van Malderen, R.: Performance assessment of the IASI-O3 KOPRA product for observing midlatitude tropospheric ozone evolution

for 15 years: validation with ozone sondes and consistency of the three IASI instruments, EGUsphere [preprint], https://doi.org/10.5194/egusphere-2024-4096, 2025.

Van Malderen, R., Thompson, A. M., Kollonige, D. E., Stauffer, R. M., Smit, H. G. J., Maillard Barras, E., Vigouroux, C., Petropavlovskikh, I., Leblanc, T., Thouret, V., Wolff, P., Effertz, P., Tarasick, D. W., Poyraz, D., Ancellet, G., De Backer, M.-R., Evan, S., Flood, V., Frey, M. M., Hannigan, J. W., Hernandez, J. L., Iarlori, M., Johnson, B. J., Jones, N., Kivi, R., Mahieu, E., McConville, G., Müller, K., Nagahama, T., Notholt, J., Piters, A., Prats, N., Querel, R., Smale, D., Steinbrecht, W., Strong, K., and Sussmann, R.: Global Ground-based Tropospheric Ozone Measurements: Reference Data and Individual Site Trends (2000–2022) from the TOAR-II/HEGIFTOM Project, EGUsphere [preprint], https://doi.org/10.5194/egusphere-2024-3736, 2025.

Kulawik, S. S., Bowman, K. W., Luo, M., Rodgers, C. D., and Jourdain, L.: Impact of nonlinearity on changing the a priori of trace gas profile estimates from the Tropospheric Emission Spectrometer (TES), Atmos. Chem. Phys., 8, 3081–3092, https://doi.org/10.5194/acp-8-3081-2008, 2008.

---

## Author Comment (AC5)

SUMMARY COMMENTS AND SUGGESTIONS FOR REVISION

This Comment follows that of HEGIFTOM Co-Chair R. Van Malderen and amplifies his major points. We refer to the same "Van Malderen et al, in press, 2025" that he does and designate it as "HEGIFTOM-1" because there is a second HEGIFTOM paper in review in the TOAR II collection. We go beyond Van Malderen's comments to 3 summary recommendations:

- The selection and number of ground-based stations for which ozone data are compared to the IASI-CDR product is inadequate in number and 'non-homogenized' datasets should not be used. In contrast to the 7 stations used in this paper, more than 25 stations, homogenized over the 2008-2019 period, need to be used in evaluation of the IASI products and the trends. The 7 sites included no equatorial stations and far too few northern mid-latitude stations, in some cases where more than sonde data are available. The recommended ozonesonde 27 stations appear in Table 1 at the end of this Comment.
- The authors conclude with general, not well-defined speculation on why there is little progress in how computed trends from the new IASI products diverge from the UV-based product trends from the Gaudel et al. (2018) TOAR I paper. More analysis and insights on this issue are needed before the paper is worthy of publication in a quality Copernicus journal. Specific questions are raised for consideration.
- HEGIFTOM-1 is now "THE Reference dataset" for trends comparisons. The mismatch of years (2008-2019 in Boynard et al. vs 2000-2022 in HEGIFTOM-1) is speculated as one reason for why the trends in this paper differ from the ground-based trends in HEGIFTOM-1. We reran the trends in HEGIFTOM-1 for 27 stations for 2008-2019, see Table below, to support a valid comparison. Some of the HEGIFTOM-1 site trend signs changed and uncertainties increased, leading to a clear TOAR-worthy conclusion that trends computed from 12 or 16 years of IASI or ground-based data is inadequate. A revision must include this important result!

Thank you for your kind and constructive feedback. We have carefully considered your comments and made the necessary clarifications and corrections in the revised manuscript. Please find a point-by-point answer below.

SYNOPSIS – This study presents an IASI product over 16 years, consisting of contributions from METOP-A, METOP-B and METOP-C, merged to create the IASI-CDR (Climate Data Record, 2008-2023). Several tropospheric ozone columns (to 450, to tropopause) are presented and compared on a monthly mean basis with (1) the CrIS IR ozone product and with (2) comparable ozonesonde columns from 43 stations over the period. Trends for global mid-latitudes and the tropics are computed using Quantile Regression (QR) for the period 2008-2019 (pre-COVID) and 2008-2023; the latter trends reflect an apparent COVID impact. As in the TOAR I paper in which Gaudel et al. (2018) summarized satellite product trends (2005/2008 – 2016) showing IASI (FORLI version) to be an outlier compared to UV-type satellite records, it appears that the IASI-CDR trends (now 2008-2019, omitting COVID period) is an outlier with the corresponding UV-based time-series. In both cases the greatest discrepancies are in the tropics except for SE Asia where all products display increases of ~1-2 DU/decade. The UV-type satellite products tend to be more variable with regions in the tropical Americas, Africa and Atlantic showing some increases (~2005-2019; Gaudel et al., 2024; Thompson et al., 2024). The IASI-CDR TOAR II-period (Fig. 12 in paper) shows little increase over Europe, a decrease over North America and only modest increases in east Asia, again in disagreement with more variable UV-type satellite trends. IASI 2008-2019 trend comparisons are made with 7 ozonesonde time-series: one subtropical site, 4 northern mid-latitude (majority European), 1 southern mid-latitude (Fig. 16). The authors speculate briefly on causes of the persistent discrepancy between the IASI vs UV trends and what is emerging as the prevailing view of tropospheric changes over the prior ~15 years. Comparisons among comparable IR products eg. IASI-CDR, IASI-KOPRA, and CrIS global ozone products are described with varying degrees of detail or with Figures that are suggestive but not conclusive.

OVERALL COMMENT – The paper poses good questions, presents a reasonable approach for its calculations and selection of results and is well-arranged. However, it leaves too many unanswered questions and does not advance the scientific understanding of its ozone trends beyond the first TOAR study. The three most important aspects of the paper that require additional analysis are summarized as follows:

1>> Quality assurance and evaluation of the IASI-CDR. There is reference to a larger set of sonde stations used for comparisons (43 stations; Fig 8 and 10) but Table 2 is inaccurate. The following stations are not available on HEGIFTOM archive because they are not homogenized datasets: Lindenberg, Prague, Tateno, Hong Kong, Broadmeadows, and Macquarie. We recommend using only homogenized datasets for reference datasets and trends. Comparison of trends is restricted to 7 stations, none truly tropical, 15 degrees or less, despite the fact that a number of cited and other TOAR II studies, both satellite and ground-based, focus on the tropics e.g., Froidevaux

et al., 2025; Gaudel et al., 2024; Thompson et al., 2025. Time-series comparisons in Fig. 16 show only 7 stations with limited geographical coverage.

Our initial selection criteria may appear overly restrictive, especially when compared to the HEGIFTOM analysis (Van Malderen et al., 2025), which included 34 homogenized stations using a more relaxed threshold of at least two monthly launches and >50% sampling.

In our original analysis, we applied more stringent selection criteria (at least three launches per month and 70% time series completeness between 2008 and 2023). These thresholds, adapted from Lu et al. (2019), were chosen to ensure the reliability of trend estimates by minimizing uncertainty in monthly means and maintaining sufficient temporal overlap with satellite retrievals.

However, in response to your comment, we repeated the trend analysis using the less restrictive criteria of at least two monthly launches and >50% sampling, consistent with the HEGIFTOM approach. This adjustment significantly increased the number of eligible stations from 7 to 27.

Importantly, for six of the seven stations included in our initial analysis, trend results remained consistent under the relaxed criteria. For one station, the trend became statistically significant with high certainty, likely due to the more complete time series provided by the less restrictive thresholds. Despite this exception, the overall agreement between the two approaches suggests that relaxing the selection criteria does not substantially affect trend outcomes. We have therefore adopted the relaxed criteria in the revised manuscript to broaden the station sample while ensuring consistent and reliable results.

Following your recommendation, we have also restricted the trend analysis to homogenized ozonesonde records only. Non-homogenized stations have been excluded from trend calculations to ensure consistency. These stations are still utilized for the validation of IASI data but are not included in the long-term trend estimation. This approach ensures a more rigorous and consistent assessment of trends.

Concerning the evaluation of IASI-CDR:

- Extensive discussion of CDR vs earlier FORLI product appears – but no illustrations of why you expect the CDR product to perform better than the earlier one.

While we did not explicitly claim that the IASI-CDR product performs better in all respects compared to the earlier FORLI product, the key improvement lies in its homogeneity and consistency over time, making it more suitable for long-term trend analysis. As shown in Boynard et al. (2018), the IASI-FORLI dataset suffers from inhomogeneities and artificial drifts caused by changes in processing versions. The IASI-CDR product was developed specifically to mitigate these artifacts by reprocessing radiances and auxiliary data consistently over the full record.

To assess the drift of IASI ozone measurements, we examined two sets of criteria:
- Original criteria (criteria 1): at least 3 soundings per month, 70% data coverage (from initial study)
- Updated criteria (criteria 2): at least 2 soundings per month, 50% data coverage (based Van Malderen et al. (2025) selection criteria).

The drift shifts from -1% per decade with criteria 1 to -2.5% per decade with criteria 2. While this change is noticeable, both estimates remain well below the 3% per decade threshold, indicating that the drift is still within an acceptable range.

The increased drift under Criteria 2 is likely due to the inclusion of more stations, which could introduce variability, especially at stations with gaps or incomplete time series. When the stricter Criteria 1 (3 soundings per month and 70% coverage) is applied, the drift returns to -1% per decade, which highlights how the drift estimate is sensitive to the selection criteria.

In contrast, the earlier FORLI product exhibited significant artificial drifts, particularly around 2010, due to changes in processing versions (Boynard et al., 2018).

These results highlight the improved consistency and homogeneity of the IASI-CDR product, making it a more reliable choice for long-term trend analysis compared to the earlier FORLI version.

- IASI-KOPRA product mentioned but why is there no extensive comparison with this product or at least a paragraph comparing results of Dufour et al. (https://egusphere.copernicus.org/preprints/2025/egusphere-2024-4096/) to those shown here, particularly for sonde comparisons?

Thank you for this suggestion. A more detailed comparison with Dufour et al. (2025) who analyzed IASI-O$_3$ KOPRA data from 2008 to 2022 using similar regional definitions is indeed relevant to contextualize our results.

For the surface-tropopause column, we find a negative trend with high certainty over Europe (–0.07 ± 0.07 DU/yr, p = 0.03), which aligns well with the trend reported by Dufour et al. (–0.05 ± 0.02 DU/yr, p = 0.03), also classified as high certainty. Over North America and Asia, both studies detect a negative trend, with medium to very low certainty

At the station level, we observe a consistent picture between the two studies. For the stations common to both datasets, trends are negative and statistically significant in all cases except Uccle, where both studies find a non-significant positive trend. The only notable discrepancy is at Boulder: Dufour et al. (2025) report a significant negative trend, while we find a non-significant small positive trend. This divergence may reflect differences in the retrieval sensitivity or sampling characteristics between the two IASI products at this high-altitude site.

Overall, both studies consistently depict negative trends in Europe with high certainty and a lack of evidence for trends across Asia and North America.

We have included this discussion in the revised manuscript.

- Vertical discrepancies mentioned in comparisons with sondes (tropical, mid-latitudes, and polar) appear in Fig. 10. Although overall IASI column amounts are compared favorably to the sondes (Line 22, only 2% offset in tropics) can this be due to a cancelling of offsets illustrated in the Figure? Discuss the potential impact on the trends.

The vertical discrepancies observed between IASI and ozonesonde profiles not inherently bias trend estimates, provided these vertical offsets remain stable over time. Our drift analysis of the tropospheric column reveals small temporal drift during the study period, indicating that the magnitude and vertical distribution of these biases have remained reasonably constant. Therefore, while vertical compensation explains the small differences in absolute column amounts, it is unlikely to distort the long-term tropospheric ozone trends derived from IASI.

- The CrIS-TROPESS a priori looks so much better than the IASI (varies with season and latitude). Although the IASI climatology (Fig. 7) looks reasonable, can the IASI a priori (Fig. B1) – with apparently little seasonal information and only latitude dependence- be a cause of the discrepancies with other products? Does inadequate representation of seasonality (monthly variations in trends are typical and significant in the tropics, for example: Stauffer et al., 2024; Thompson et al., 2021) propagate to trends that disagree with both sondes and UV-products? What additional insights can you derive from scatterplots with ozonesondes (Figs. A1-4)? Comparing to sondes seasonally can help identify discrepancies. There is extensive discussion of similarities and differences with CrIS but it doesn't get to the crux of understanding the large negative ozone trends here that are at odds with other products.

We thank the reviewer for this insightful comment. While the IASI retrievals use a climatological a priori profile (latitude-dependent, non-seasonal), our results show good agreement with CrIS-TROPESS, which uses a spatially and seasonally varying a priori. This suggests that the simplified IASI a priori is not the dominant driver of discrepancies with other products.

We have conducted an additional analysis to investigate this point more deeply. Specifically, we applied the IASI a priori (fixed in latitude, not seasonally resolved) to the TROPESS retrieval algorithm, which normally uses a spatially and seasonally varying a priori. The results indicate that the choice of a priori profile clearly influences the retrieved ozone profiles. However, the differences observed between the IASI and TROPESS products are smaller than the differences between their respective a priori profiles. This is further illustrated by Figure 1 showing that the difference between the priors (right maps) is larger than the difference between the retrieved TROPESS and IASI columns (left maps). Additionally, the difference between the priors (right maps) is also larger than the difference observed when swapping the priors (middle maps). While the left and middle maps have comparable magnitudes, their spatial distributions differ to some extent.

These findings suggest that the a priori alone does not fully explain the discrepancies between the datasets. Moreover, the retrieval process is not strictly linear with respect to the a priori assumptions, indicating that a complete retrieval using the IASI a priori would be necessary to accurately quantify its impact (Kulawik et al., 2008).

We have revised the manuscript to include this analysis and discussion, emphasizing that differences in retrieval methodologies, such as the treatment of prior covariance matrices and the representation of vertical ozone profiles, likely contribute significantly to the observed inconsistencies.

[Figure]

Figure 1. Spatial distribution of tropospheric ozone (Dobson Units) over the period 2016-2022: left: difference between CrIS (TROPESS) and IASI retrievals; middle: difference between CrIS retrievals using the IASI and TROPESS a priori; right: difference between the IASI and TROPESS a priori.

Figures A1–A4 present scatterplots between IASI and ozonesonde data for four vertically integrated subcolumns (surface–300 hPa, 300–150 hPa, 150–25 hPa, and 25–3 hPa), chosen to reflect the vertical sensitivity of the retrieval. These comparisons, which aim to validate the four subcolumns defined in Boynard et al. (2018) with the maximum sensitivity, show overall consistency (with biases typically <20%). However, these comparisons are not broken down by season. We agree that a seasonal comparison could provide further insight into discrepancies and trend behavior, and this will be explored in future work.

Finally, we note that our trend results align well with those from Dufour et al. (2023), reinforcing confidence in the robustness of our approach.

2>>  Inadequate number of ground-based (GB) reference sites.  There are two aspects of this issue.

First, the authors show 43 potential stations (Table 2, Fig. 8) but make trend comparisons with only 7 ozonesonde data sets. Perhaps they don't understand the fine points of which data are appropriate to use (see Van Malderen comment). There is no need to speculate, as the authors have done, on why their results do not resemble those from mostly UV sensors. All satellites now have the HEGIFTOM-1 trends at individual stations (some with

multiple instruments, not only sondes) as the gold standard independent reference at 55 sites total for 2000-2022. The HEGIFTOM reprocessing includes references of the data for each instrument type (sonde, FTIR, UV Umkehr) to a global absolute standard. Your trends analysis should add at least 20 stations (exclude polar sites where IASI-CDR struggles with DOFS) to have a more representative picture of IASI performance. Note that of the 7 reference sites (Boulder, Hilo, Lauder), there is more than one GB record for comparison. The stations in the Table below have sufficiently temporally dense records for comparison.

Second, Line 330 states that months with only 1-2 sondes/month give 'inadequate' results for 50%-ile trends. In the accepted version of HEGIFTOM-1, it is shown that these trends (computed with QR or MLR) are unaffected by cutting from 4-5 sondes/ month to 2; only the uncertainty changes (increases). That is a second justification for using more sonde locations for the GB comparisons - candidates in the Table.

Please see answer above, which indicates that in the revised manuscript, 27 stations are used for the assessment of the trends.

3>> Brevity of the IASI record is a concern for 2008-2019 trends. For comparison, we re-ran the QR trends for the authors' selected 7 stations as well as 20 other HEGIFTOM reference ozonesonde stations (excluding polar sites). The results are listed in the Table below, which also includes results from the HEGIFTOM-1 2000-2022 trends for surface-300 hPa (in ppbv/decade) as well as XO3 (ppbv/decade) and DU (DU/decade) for surface to tropopause tropospheric columns as a reference.

Note for the recalculated 2008-2019 ozonesonde trends, that 4 out of the 7 author selected stations change sign of trend from the longer time series to the shorter time series. For example, Boulder, a station with a high certainty ($p < 0.05$) associated with a negative trend for the longer time series (2000-2022), has a slightly positive trend with large uncertainties for the 2008-2019 period. Of the 27 ozonesonde stations listed in the table, 9 sites have trend sign changes. Discussion on the point of reduced reliability and value of the shorter time series (12 or 16 years vs 23 years in HEGIFTOM and sonde studies*) is needed and is now a view that can be made with confidence. In a sort of "reversal" of your paper's message, a significant advance and outcome of your paper, with the contracted (2008-2019) HEGIFTOM calculation, is that TOAR II needs to recognize the limitations of datasets that cover fewer than ~20 years!

The uncertainties in the trends also increase with the shorter time series (ie. double those for the 2000-2022 time period – see Table below).

Thank you for highlighting the critical role of time series length in trend analysis. We fully agree that longer records, such as the 23-year HEGIFTOM dataset (2000–2022), offer more robust and stable trend estimates with reduced sensitivity to interannual variability.

Our primary trend analysis is based on the 16-year period from 2008–2023, which we consider sufficiently long to yield meaningful and statistically robust results. The shorter 2008–2019 window was included specifically to examine potential short-term impacts of the COVID-19 pandemic. As expected, trends over this shorter period show larger uncertainties and greater sensitivity to variability, including sign changes, as also observed in your HEGIFTOM recalculations.

To assess sensitivity to time series length, we expanded our analysis to 27 ozonesonde stations and compared trends for both 2008–2019 and 2008–2023. Results show that extending the record length significantly reduces trend uncertainties across most stations.

While we agree that datasets shorter than ~20 years must be interpreted with caution, especially in isolation, we believe that 16 years strikes a reasonable balance between capturing recent variability and ensuring trend reliability. Moreover, the general consistency between IASI trends and those from ozonesondes and other studies (e.g., Dufour et al. 2025) lends confidence to our findings.

We appreciate your observation that our results underscore the limitations of shorter records, a conclusion we fully support and that further motivates the need for sustained long-term observations.

On your paper (also noted by R. Van Malderen) the reported uncertainties in Fig. 16 seem small for only 12 years of data. Can you check those and discuss your bootstrap method in more detail?

Trend calculations follow the guidelines established by the TOAR-II Statistics Focus Working Group (Chang et al., 2023b). Specifically, trends are estimated using quantile regression (QR) at the 50th percentile, and a moving

block bootstrap method is used to assess the uncertainty of the derived trends and to calculate p-values for evaluating their statistical significance. We used the toarstats Python package (https://gitlab.jsc.fz-juelich.de/esde/toar-public/toarstats). The monthly ozone column time series are first deseasonalized by fitting and removing a sine-cosine model with 12- and 6-month periodicities to eliminate seasonal variations. The relatively small uncertainty ranges reported in the manuscript correspond to uncertainties expressed as ±1σ. To improve clarity and consistency, we have revised the manuscript accordingly and now report uncertainties as ±2σ.

*In addition to the sondes studies you have referenced, Stauffer et al., 2024; Thompson et al., 2021; Thompson et al., 2024, there is an excellent new sonde trends paper submitted on Réunion SHADOZ and SAOZ time-series (1998-2021) submitted to Earth and Space Science: https://essopenarchive.org/doi/full/10.22541/essoar.174594999.98715985/v1

'HEGIFTOM-1" below is posted. Final version is in press.
Van Malderen, R., Thompson, A. M., Kollonige, D. E., Stauffer, R. M., Smit, H. G. J., Maillard Barras, E., Vigouroux, C., Petropavlovskikh, I., Leblanc, T., Thouret, V., Wolff, P., Effertz, P., Tarasick, D. W., Poyraz, D., Ancellet, G., De Backer, M.-R., Evan, S., Flood, V., Frey, M. M., Hannigan, J. W., Hernandez, J. L., Iarlori, M., Johnson, B. J., Jones, N., Kivi, R., Mahieu, E., McConville, G., Müller, K., Nagahama, T., Notholt, J., Piters, A., Prats, N., Querel, R., Smale, D., Steinbrecht, W., Strong, K., and Sussmann, R.: Global Ground-based Tropospheric Ozone Measurements: Reference Data and Individual Site Trends (2000–2022) from the TOAR-II/HEGIFTOM Project, EGUsphere [preprint], https://doi.org/10.5194/egusphere-2024-3736, 2025.

Also referred to is Thompson et al., submitted, 2024, Posted in 2025:
Thompson, A. M., Stauffer, R. M., Kollonige, D. E., Ziemke, J. R., Cazorla, M., Wolff, P., and Sauvage, B.: Tropical Ozone Trends (1998 to 2023): A Synthesis from SHADOZ, IAGOS and OMI/MLS Observations, EGUsphere [preprint], https://doi.org/10.5194/egusphere-2024-3761, 2025.

TABLE. This is essentially an update of Table 1 in HEGIFTOM-1, Van Malderen et al., in press, 2025, run again with the QR method, same as employed in Boynard et al. Ozonesonde stations with homogenized data and sufficient sample size are listed and exclude near-polar regions. Yellow-coded lines represent the 7 author-selected IASI-sonde comparison stations. Orange-coded lines indicate where the sign of the trend changes based on the different periods of trends calculations (23 years vs. 12 years).

TABLE IS POSTED IN SUPPLEMENT

Citation: https://doi.org/10.5194/egusphere-2025-1054-CC3 IASI-Boynard Paper Comment – 13 May 2025
Anne Thompson, Debra Kollonige, Ryan Stauffer

References
Boynard, A., Hurtmans, D., Garane, K., Goutail, F., Hadji-Lazaro, J., Koukouli, M. E., Wespes, C., Vigouroux, C., Keppens, A., Pommereau, J.-P., Pazmino, A., Balis, D., Loyola, D., Valks, P., Sussmann, R., Smale, D., Coheur, P.-F., and Clerbaux, C.: Validation of the IASI FORLI/EUMETSAT ozone products using satellite (GOME-2), ground-based (Brewer–Dobson, SAOZ, FTIR) and ozonesonde measurements, Atmos. Meas. Tech., 11, 5125–5152, https://doi.org/10.5194/amt-11-5125-2018, 2018.

Kulawik, S. S., Bowman, K. W., Luo, M., Rodgers, C. D., and Jourdain, L.: Impact of nonlinearity on changing the a priori of trace gas profile estimates from the Tropospheric Emission Spectrometer (TES), Atmos. Chem. Phys., 8, 3081–3092, https://doi.org/10.5194/acp-8-3081-2008, 2008